# Tuning the activity and selectivity of electroreduction of $CO_2$ to synthesis gas using bimetallic catalysts

Ji Hoon Lee[1,6], Shyam Kattel[2,6], Zhao Jiang[3], Zhenhua Xie[1,4], Siyu Yao[4], Brian M. Tackett[1], Wenqian Xu[5], Nebojsa S. Marinkovic[1,4] & Jingguang G. Chen[1,4]

The electrochemical carbon dioxide reduction reaction to syngas with controlled $CO/H_2$ ratios has been studied on Pd-based bimetallic hydrides using a combination of in situ characterization and density functional theory calculations. When compared with pure Pd hydride, the bimetallic Pd hydride formation occurs at more negative potentials for Pd-Ag, Pd-Cu, and Pd-Ni. Theoretical calculations show that the choice of the second metal has a more significant effect on the adsorption strength of *H than *HOCO, with the free energies between these two key intermediates (i.e., $\Delta G(*H)-\Delta G(*HOCO)$) correlating well with the carbon dioxide reduction reaction activity and selectivity observed in the experiments, and thus can be used as a descriptor to search for other bimetallic catalysts. The results also demonstrate the possibility of alloying Pd with non-precious transition metals to promote the electrochemical conversion of $CO_2$ to syngas.

[1] Department of Chemical Engineering, Columbia University, New York, NY 10027, USA. [2] Department of Physics, Florida A&M University, Tallahassee, FL 32307, USA. [3] Department of Chemical Engineering, School of Chemical Engineering and Technology, Xi'an Jiaotong University, No.28 Xianning West Road, 710049 Xi'an, Shaanxi, P. R. China. [4] Chemistry Division, Brookhaven National Laboratory, Upton, NY 11973, USA. [5] X-ray Science Division, Advanced Photon Source, Argonne National Laboratory, 9700 South Cass Ave, B433/D003, Argonne, IL 60439, USA. [6] These authors contribute equally: Ji Hoon Lee, Shyam Kattel. Correspondence and requests for materials should be addressed to J.G.C. (email: jgchen@columbia.edu)

The utilization of fossil fuels has increased the atmospheric $CO_2$ level, giving rise to serious environmental concerns[1]. Therefore, many research endeavors have been focused on renewable energy sources, such as wind and solar. However, it is often difficult to merge these resources directly into the current electricity network due to their intermittent availabilities. As an effort to overcome this challenge, electrochemical carbon dioxide ($CO_2$) reduction reaction ($CO_2$RR), coupled with such renewable energy resources, has drawn significant attention and is regarded as one of the viable options for a future sustainable energy economy[2–6].

Among the various $CO_2$RR pathways[6–9], the simultaneous production of CO and $H_2$ (syngas) with tunable CO/$H_2$ ratios has been considered very beneficial, because the obtained syngas can be subsequently utilized for producing value-added chemicals through existing thermochemical processes, such as the Fisher–Tropsch and methanol synthesis reactions[2,10–12]. For these down-stream thermocatalytic processes, the CO/$H_2$ ratio plays an important role in controlling the product selectivity. Therefore, the ability to produce syngas with a controlled CO/$H_2$ ratio should provide versatility in a hybrid electrocatalysis/thermocatalysis approach for $CO_2$ utilization.

It is not a trivial task to achieve a high $CO_2$RR activity while maintaining a CO/$H_2$ ratio in the desirable range for thermochemical processes, typically between 0.5 and 2.0. Most of the $CO_2$ electrocatalysts, such as gold (Au)[13–18], silver (Ag)[10,19,20], and copper (Cu)[21–24], could not produce the desired CO/$H_2$ ratio with high $CO_2$RR activity. In this study, we overcome this challenging issue by using palladium (Pd)-based bimetallic hydrides for electrochemical syngas production with controlled CO/$H_2$ ratios. Pd itself is among the traditional formate-producing metals like tin (Sn) and lead (Pb)[6,25–27]. However, once palladium hydride (PdH) forms via the cathodic reactions[28], it is possible to produce the mixture of CO and $H_2$ as the major product of $CO_2$RR[2,29–31]. We compared a series of Pd–M bimetallic catalysts to determine how the presence of the second metal affects the PdH formation and the $CO_2$RR performance. The combined experiments and DFT calculations reveal that the Gibbs free energy difference of *HOCO and *H, which are the key reaction intermediates for the $CO_2$RR and HER, respectively, can be a potential descriptor for predicting the $CO_2$RR activity and CO/$H_2$ ratio. Furthermore, our results demonstrate the feasibility of promoting syngas production by alloying Pd with non-precious, first-row transition metals such as Ni and Cu.

## Results

**Electrochemical $CO_2$ reduction performance for bimetallic Pd alloys.** The electrochemical $CO_2$RR was performed using a series of 10$wt$ % Pd-based bimetallic nanoparticles (NPs) supported on Vulkan carbon (C). The second metals, cobalt (Co), nickel (Ni), copper (Cu), silver (Ag), and platinum (Pt), were selected as they can make an alloy with Pd at an atomic Pd:M ratio of 8:2. This ratio was chosen in this study because PdNi in this ratio (i.e., $Pd_{80}Ni_{20}$) represented optimal composition for the PdNi catalysts with different ratios (Supplementary Fig. 1 and Supplementary Table 1). Each final sample was named as PdM, where M indicates the second metal used in this study (M = Co, Ni, Cu, Ag, and Pt). The successful synthesis of bimetallic Pd alloys was confirmed by powder X-ray diffraction pattern (XRD) analysis (Supplementary Fig. 2a), indicating that these materials kept the same face-centered cubic structure (space group: Fm-3m). The peak locations for the (111) and (200) planes were identified to follow the Vegard's law. In the cases of using M with larger atomic radius than Pd (M = Ag and Pt), those peaks shift toward lower $2\theta$ values. On the other hand, those peaks shift toward

higher $2\theta$ values in the cases of using M with smaller atomic radius (M = Cu, Ni, and Co) (Supplementary Fig. 2b).

The $CO_2$RR activity and the CO/$H_2$ ratio of these bimetallic samples were evaluated by using the chronoamperometry technique in $CO_2$-saturated 0.5 M sodium bicarbonate ($NaHCO_3$) electrolyte with vigorous magnetic stirring at different potentials (Supplementary Fig. 3). The gaseous product was analyzed using gas chromatography (See the Supplementary Methods), and CO and $H_2$ were the two major products. It is well known that Pd-based catalysts are vulnerable to CO-poisoning[2,32,33] due to its strong binding to CO. However, the transformation of Pd-to-PdH under the $CO_2$RR condition, which will be discussed later, reduces the binding energy of CO and thus enables facile CO desorption. The Faradaic efficiencies (FEs) of CO (FE(CO)) and $H_2$ (FE($H_2$)) and CO/$H_2$ ratio are plotted versus the applied potential in Supplementary Figs. 4 and 5, respectively. Overall, the total FE from CO and $H_2$ reached up to ~80 % at −0.8 V versus the reversible hydrogen electrode ($V_{RHE}$, here and onward) and ~100% after −0.9 $V_{RHE}$, consistent with CO and $H_2$ being the major products. Formic acid (HCOOH) as a minor product was the only liquid product at all of the potentials, which thus accounted for the rest of FE. Its quantification was done by using $^1$H NMR measurements (See the Supplementary Methods). The FE values of HCOOH were determined to be 5~20% from −0.6 to −0.8 $V_{RHE}$ and became negligible at −0.9 $V_{RHE}$ and thereafter. As the potential is scanned more cathodically, FE(CO) tends to initially increase, then saturate, and finally decrease while FE($H_2$) increases. This is because $CO_2$RR is controlled by the mass transport of dissolved $CO_2$, while HER is not limited by the proton transport from $H_2O$. The Tafel plots of CO and $H_2$ support this interpretation (Supplementary Fig. 6). The CO/$H_2$ ratio and partial current density of CO (J(CO)) also show the similar potential dependent profiles, as displayed in Supplementary Figs. 5 and 7, respectively. The FE(CO), CO/$H_2$ ratio, and J (CO) at −0.9 $V_{RHE}$ are shown in Fig. 1, which illustrates that the second metal component significantly modifies the $CO_2$RR activity of the PdM bimetallic catalysts. The FE(CO) and CO/$H_2$ ratio follow the trend of PdAg > PdCu > PdNi > Pd > PdCo > PdPt (Fig. 1a, b). The J(CO) also follows the same trend except for PdCo (Fig. 1c), which is attributed to its slightly higher total current density compared with the others. This trend remains the same still even with a different electrolyte (i.e., 0.5 M $KHCO_3$) while the values of J(CO) and FE(CO) are enhanced significantly due to the different hydrolysis effects[21,22] of $Na^+$ and $K^+$ (Supplementary Fig. 8). For comparison, 10 wt% Au NPs supported on C (Au/C), a benchmark electrocatalyst for selective $CO_2$-to-CO conversion with low HER activity[6], was also tested and summarized in the Supplementary Information (Supplementary Figs. 3 and 4). Under similar electrochemical conditions, Au/C produced primarily CO, with $H_2$ being a minor product (e.g., CO/$H_2$ ratio of 3.8 at −0.7 $V_{RHE}$). In contrast, PdM bimetallic catalysts such as PdAg produced both CO and $H_2$ with high activity, representing a better catalyst for syngas production than Au/C (Supplementary Note 1). The stability of the Pd and PdNi catalysts was characterized using Transmission electron microscopy (Supplementary Fig. 9), which revealed that both Pd and PdNi maintained the original particle size after $CO_2$RR. Furthermore, the elemental mapping verified that Pd and Ni were in close proximity, consistent with the formation of PdNi alloy both before and after $CO_2$RR (Supplementary Fig. 10).

**In situ X-ray absorption fine structure analysis.** In order to investigate the local environment variations around Pd in the PdM bimetallic electrocatalysts during $CO_2$RR, in situ XAFS analysis was performed (Fig. 2 and Supplementary Figs. 11–14,

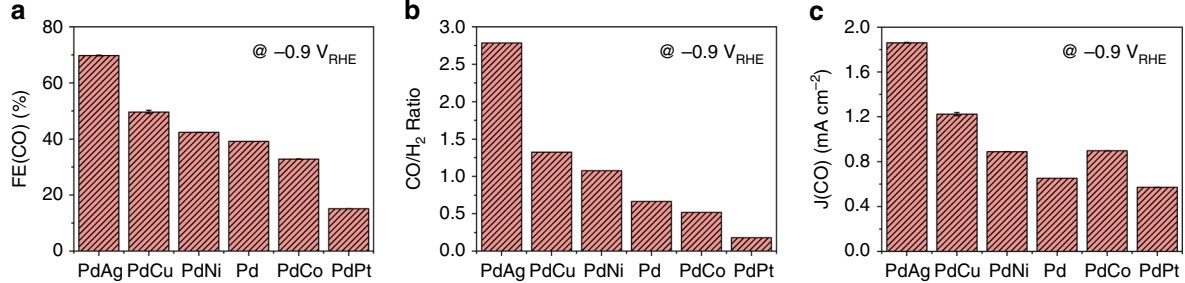

**Fig. 1** Electrocatalytic performance of different catalysts at $-0.9$ $V_{RHE}$. **a** Faradaic efficiency (FE(CO)) of CO. **b** CO/$H_2$ ratio. **c** Partial current density (J (CO)) of CO

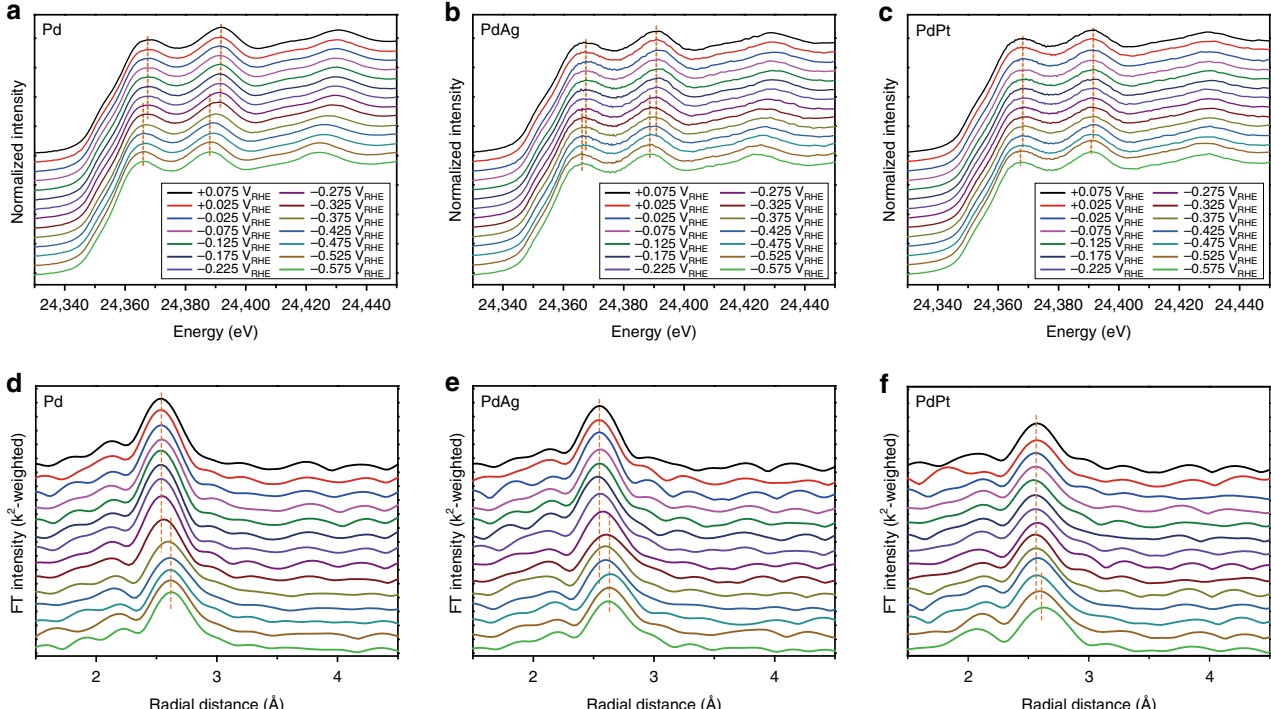

**Fig. 2** In situ XAFS profiles measured at Pd K-edge. **a–c** Normalized XANES profiles for **a** Pd, **b** PdAg, and **c** PdPt. **d–f** $k^2$-weighted EXAFS spectra for **d** Pd, **e** PdAg, and **f** PdPt

see the Methods). The XAFS analysis provides useful information, especially for Pd-based materials, because the hydride formation proceeds concurrently with the reduction of $Pd^0$-to-$Pd^{-1}$ and the increase in Pd-Pd distance[2,28,34,35]. X-ray absorption near edge structure (XANES) analyses at the Pd K-edge indicate that, during $CO_2RR$, the oxidation state of Pd progressed to $-1$ for Pd and PdAg (Fig. 2a, b), implying that the presence of Ag in the Pd lattice did not interfere with the formation of the PdH phase. Likewise, PdCu also shows the phase transition to PdCu hydride under the same condition (Supplementary Fig. 11). On the other hand, the XANES profile for PdPt (Fig. 2c) shows that the oxidation state of Pd remains almost unchanged throughout the entire potential sweeping. This might be because Pt in the PdPt alloy has a strong *H binding, and consequently, the formation of PdPt hydride phase (i.e., H diffusion into lattice) is no longer favorable. Such an interpretation is also consistent with the recent report on the inhibitive role of Pt on PdH formation[34]. The reduction of Pd is visualized more clearly in the Pd K-edge energy ($E_0$) variation versus the applied potentials (Supplementary Fig. 12). While the $E_0$ values of Pd, PdAg, and PdCu decreased by

1 eV around $-0.325$ $V_{RHE}$, that of PdPt maintained its original value until $-0.525$ $V_{RHE}$.

The extended XAFS (EXAFS) analyses (Fig. 2d–f and See the Supplementary Methods) provide information on the structural variations during the PdH phase formation. The EXAFS fitting profiles are shown in Supplementary Fig. 13 and the fitting parameters are tabulated in Supplementary Tables 2–10. The EXAFS analyses reveal that the distance of Pd-to-Pd/M (M = Ag and Cu) increased gradually from ~2.75 (at 0.075 $V_{RHE}$) to ~2.82 Å (at $-0.575$ $V_{RHE}$), consistent with the volume expansion as a result of hydride formation. However, PdPt exhibited negligible changes of bond length in both Pd-Pd ($2.75 \rightarrow 2.76$ Å) and Pd–Pt ($2.71 \rightarrow 2.74$ Å) distances, confirming that the bimetallic PdPt hydride formation is inhibited in the presence of Pt in the PdPt alloy[34]. From the combined results of XAFS and electrochemical measurements, the formation of PdH phase is likely a key step for the simultaneous CO and $H_2$ production over Pd-based electrocatalysts.

For reference, in situ XANES analyses were also conducted at the Ag K-edge, Pt $L_3$-edge, and Cu K-edge (Supplementary

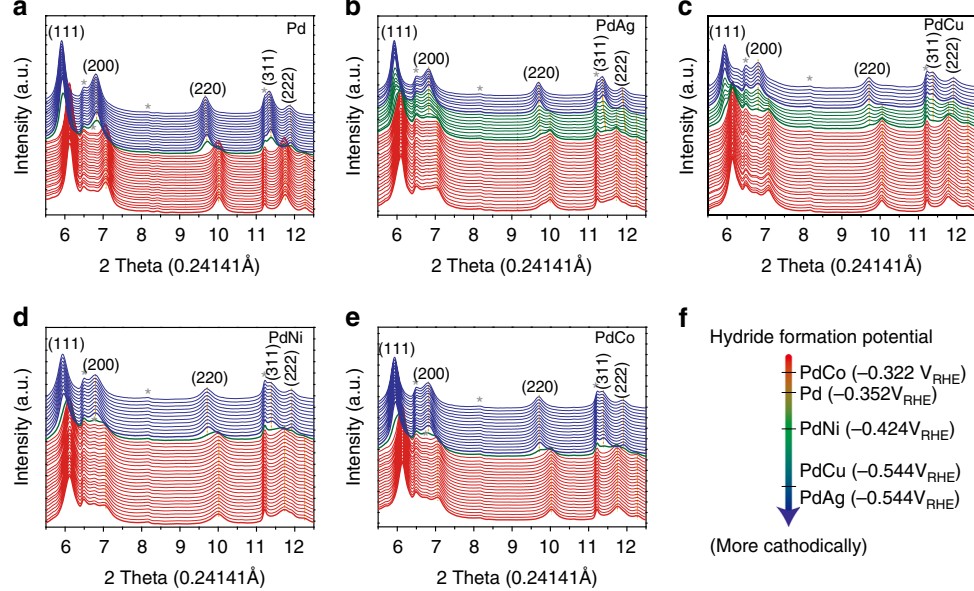

**Fig. 3** In situ XRD analyses for different catalysts. In situ XRD profiles for **a** Pd, **b** PdAg, **c** PdCu, **d** PdNi, and **e** PdCo. **f** The schematic energy diagram illustrating their potentials of palladium hydride formation. In **a–e** red and blue lines indicate Pd and PdH phases, respectively. Green lines indicate their two phase regions. Asterisk (*) marks point out the peaks from the carbon substrate

Fig. 14). The oxidation state of the second M remained unchanged through the entire $CO_2RR$ condition, suggesting that the hydride formation did not significantly change the electronic properties of the second M.

**In situ X-ray diffraction analysis.** In order to determine the potential for bimetallic PdH phase formation, in situ XRD analyses were conducted for all the bimetallic catalysts (Fig. 3 and Supplementary Figs. 15, 16). During the $CO_2RR$ under the linear sweeping voltammetry (LSV) mode, XRD patterns were obtained at different potentials from 0.2 to −0.7 $V_{RHE}$ with a scan rate of 0.05 mV s$^{-1}$. The potential interval for each XRD spectra in Fig. 3 was 24 mV, which was sufficient to capture the phase transition of Pd (red lines) to PdH (blue lines). During the $CO_2RR$, the peaks shifted toward lower $2\theta$ values due to the lattice expansion during the hydride formation (Fig. 3a–e)[2]. The onset and ending potentials for hydride formation in PdM depended on the second M. As the peaks in both Pd and PdH phases did not split into two peaks of the elemental constituents (i.e., Pd/M and PdH/M) throughout the entire in situ XRD analyses, Pd and M were randomly distributed as a solid-solution in both phases. On the other hand, in situ XRD analysis for PdPt shown in Supplementary Fig. 15 did not show the presence of hydride phase, consistent with the EXAFS results (Fig. 2, Supplementary Fig. 13, and Supplementary Tables 8–9) of the role of Pt in preventing the hydride formation[34]. Therefore, the in situ XRD results reveal that the formation of PdH can be tuned by the choice of the second M. Moreover, the tendency of (PdM)H formation potentials on the choice of the second M coincides with their syngas ratios between CO and $H_2$ (Fig. 1b). Therefore, the bimetallic PdH formation plays an important role in the syngas production with controllable $CO/H_2$ ratios.

Based on the results from the combined in situ XRD and XAFS analyses, Pd and M in the series of PdM alloys were kept mixed randomly in a solid-solution even after the bimetallic Pd hydride formation (PdM)H. However, it should be noted that the second M did not change their oxidation states during the phase transition of PdM-to-(PdM)H (Supplementary Fig. 14), suggesting that the presence of M could modify the redox potential of

$Pd^0M$ to $(Pd^{-1}M)H$ (as shown in Fig. 3f) and consequently the $CO_2RR$ performance.

The redox potential is directly related to the Fermi energy level ($E_F$) of the primary redox center[36], which should be near the empty $d$-band of Pd in this study. Therefore, the different (PdM)H potentials shown in Fig. 3f can be useful in explaining the effect of the second M on the observed $CO_2RR$ activity. In the case of M = Ag, Cu, and Ni, where $CO/H_2$ ratios were enhanced, their (PdM)H formation potentials shifted downward in comparison with PdH (−0.544 $V_{RHE}$ for (PdAg)H and (PdCu)H, −0.424 $V_{RHE}$ for (PdNi)H vs. −0.352 $V_{RHE}$ for PdH). On the other hand, (PdCo)H with the decreased $CO/H_2$ ratios showed the upward shift in the hydride formation potential by ~0.03 $V_{RHE}$. Therefore, it appears that the $CO/H_2$ ratios can be effectively controlled by the hydride formation potentials depending on the choice of the second M, as confirmed in the DFT results below.

**Density functional theory calculations.** The DFT calculations[37–45] were performed to gain insight into the effect of the second M in modifying the electronic properties of bimetallic (PdM)H and their electrocatalytic activities toward the HER and $CO_2RR$. First, the shift in the potential for PdH formation observed in in situ XRD analyses (Fig. 3) can be explained by the formation energies of PdH and (PdM)H phases. To this end, the hydride formation for Pd and PdM is assumed to occur according to the following reaction (1):

$$Pd_4 + M + 2H_2 \rightarrow (Pd_3M)H_4 + Pd \qquad (1)$$

Thus, the hydride formation energy (ΔE) is calculated as the following Eq. (2):

$$\Delta E = [E((Pd_3M)H_4) + E(Pd)] - [E(Pd_4) + E(M) + 2E(H_2)] \qquad (2)$$

where, $E(Pd_4)$ is the DFT calculated total energy of a Pd unit cell in the face-centered cubic crystal structure, $E((Pd_3M)H_4)$ is the DFT calculated total energy of the (PdM)H unit cell in the $L1_2$ crystal structure, $E(Pd/M)$ is the cohesive energy of Pd/M and $E(H_2)$ is the DFT calculated total energy of $H_2$ in the gas phase.

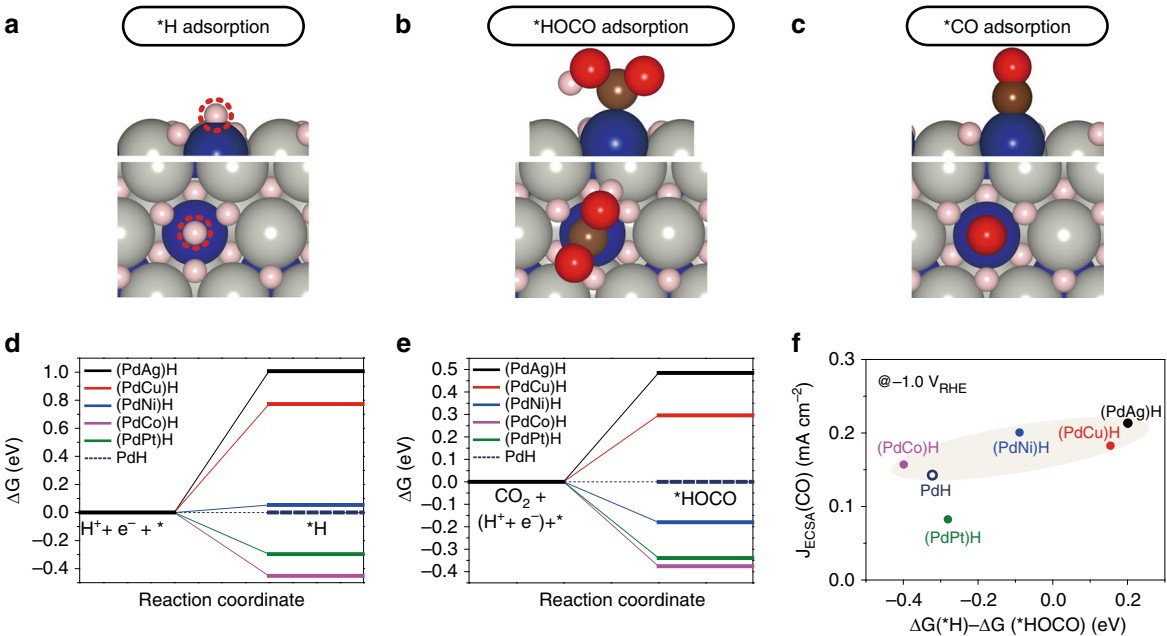

**Fig. 4** Density functional theory results. **a–c** DFT optimized configurations of **a** *H, **b** *HOCO, and **c** *CO adsorption on (PdM)H. Pd (gray), M (blue), O (red), and H (pink). *H is shown within dotted red circle in (**a**). **d**, **e** The changes in free energy (ΔG) for the rate limiting steps in **e** HER and **e** $CO_2$RR. **f** A plot of the free energy difference of *H and *HOCO adsorptions (ΔG (*H)−ΔG (*HOCO)) versus $J_{ECSA}$(CO) at −1.0 $V_{RHE}$

Supplementary Table 11 shows the DFT calculated $\Delta E$ values of bimetallic (PdM)H in the current study. It is found that reactive metals, such as Pt, Co, and Ni, which would bind H strongly, favor the (PdM)H formation compared to less reactive metals such as Ag and Cu. A positive $\Delta E$ value indicates that, from the viewpoint of experiments, a more negative potential is required for (PdM)H formation. In this regard, the DFT calculated trend in $\Delta E$ of (PdM)H (i.e. in the order of favorable hydride formation, Pt > Co > Ni > Pd > Cu > Ag) correlates relatively well with the experimentally observed positive potential shift in (PdM)H formation (PdCo > Pd > PdNi > PdCu ≈ PdAg). The main exception is for the case of PdPt, which is again attributed to the inhibitive role of Pt on hydride formation.

Additional DFT calculations[14] were performed to determine the binding energies of the HER intermediate (i.e. *H) and $CO_2$RR intermediates (i.e. *HOCO and *CO) over PdH and (PdM)H surfaces (Fig. 4a–c and Supplementary Table 12). The particle size for all samples is in the range from 5 to 10 nm (Supplementary Fig. 17), thus enabling us to use their (111) surface as a platform for further calculations. The binding energies of intermediates calculated on second M sites are used in the DFT discussion to study the immediate effect on HER and $CO_2$RR due to the presence of second M on the bimetallic catalyst surfaces (Supplementary Table 13). The DFT calculations reveal a correlation between *H and *HOCO binding energies (Supplementary Fig. 18). Both *H and *HOCO adsorption steps are difficult on (PdAg)H and (PdCu)H compared to PdH. On the other hand, (PdCo)H and (PdPt)H are predicted to show favorable adsorption of *H and *HOCO compared to PdH. For (PdNi)H, *H adsorption is slightly unfavorable while *HOCO adsorption is slightly favorable compared to PdH. Overall, the choice of the second M modified the adsorption energy of *H more than that of *HOCO (Fig. 4d, e).

The free energy (ΔG) diagrams for the HER and $CO_2$RR in Supplementary Fig. 19 were then calculated using the binding energies of reaction intermediates with their most stable adsorption configurations as displayed in Fig. 4a–c. As an effort to find a key descriptor for the $CO_2$RR with controlled CO/$H_2$

ratios, ΔG of each reaction intermediate was first plotted vs. the current density of the corresponding products $J_{ESCA}$(CO) and $J_{ECSA}$($H_2$) normalized by the electrochemical surface area (ECSA) in Supplementary Fig. 20 and Supplementary Fig. 21 (See the Methods and Supplementary Table 14). It was noted that, while ΔG(*H) could be linearly correlated with $J_{ECSA}$($H_2$), $J_{ECSA}$(CO) could not be scaled with any of the individual value of ΔG (*H), ΔG(*HOCO) or ΔG(*CO).

The ΔG diagrams in Supplementary Fig. 19 show that the *H and *HOCO adsorption is the rate limiting step for HER and $CO_2$RR, respectively, suggesting that their free energies of adsorption should play an important role in determining the CO/$H_2$ ratio in the product. Therefore, the combined effects on *H and *HOCO adsorption most likely determine the reaction pathways over (PdM)H between the HER and $CO_2$RR. As shown in Fig. 4f, the free energy difference between *H and *HOCO, i.e., ΔG(*H)−ΔG(*HOCO), correlates well with the trends in the $CO_2$RR activity (i.e., $J_{ECSA}$(CO) at −1.0 $V_{RHE}$) observed in experiments. Note again that PdPt does not follow the trend due to the inhibitive role of Pt for hydride formation[34]. However, the same consideration using *H or *CO adsorption alone does not correlate with $J_{ECSA}$(CO) or $J_{ECSA}$($H_2$) as displayed in Supplementary Figs. 22 and 23. The binding energy of O has been predicted as a potential descriptor of $CO_2$RR[46,47]. However, in the current study O reacts with surface H to form OH on (PdM)H. The DFT calculated OH formation energy does not correlate with the experimental $J_{ECSA}$(CO) or $J_{ECSA}$($H_2$) (Supplementary Fig. 24 and Supplementary Table 15), suggesting that the binding energy of O/OH is not a descriptor for $CO_2$RR over the (PdM)H catalysts. Similarly, a previous study[48] did not observe a consistent trend between ΔG(*H)−ΔG(*CO) and J(CO)/J($H_2$) of Cu-deposited Au. Thus, the combined experimental observations and DFT calculations indicate that ΔG(*H)−ΔG(*HOCO) can be used as a potential descriptor of $CO_2$RR activities on Pd-based bimetallic hydrides. Hence, the $CO_2$RR would be accelerated when ΔG(*H)−ΔG(*HOCO) is positive (e.g. on (PdAg)H and (PdCu)H), due to destabilized *H adsorption and/or stabilized *HOCO adsorption while the $CO_2$RR would be

decelerated when $\Delta G(*H)−\Delta G(*HOCO)$ is negative (e.g. on (PdCo)H) due to stabilized *H binding and an enhanced HER and/or destabilized *HOCO adsorption. The $\Delta G$ values of the *HCOO species, which is a key intermediate for HCOOH formation, over PdH and (PdM)H were calculated (Supplementary Table 16). For those catalysts that are favorable for CO production, the formation of the *HCOO intermediate is slightly favored over *HOCO. However, considering the low yield of formic acid at high overpotentials, the DFT results suggest that the production of CO could also be potentially promoted from the *HCOO pathway. While more detailed study using in situ infrared and Raman spectroscopies will be needed to further characterize the surface intermediates, the DFT results in Supplementary Table 16 suggest that both *HCOO and *HOCO intermediates potentially lead to CO production. Additional DFT calculations performed on the (100) surfaces of PdH and (PdNi) H show a similar trend in BE(*H)−BE(*HOCO) compared to the corresponding (111) surfaces (Supplementary Table 17). Thus, the (111) surface used in DFT modeling is a reasonable representation in identifying trends of relatively large nanoparticles.

## Discussion

In summary, we have demonstrated the modifying effect of alloying Pd with a second M on the phase transition of bimetallic Pd-to-PdH, leading to systematic trends in the $CO_2RR$ to syngas with controlled $CO/H_2$ ratios. The combined electrochemical evaluation, in situ characterization and DFT calculations reveal that the difference of $\Delta G$ of *H and *HOCO (i.e., $\Delta G(*H)−\Delta G(*HOCO)$) is a potential descriptor for $CO_2RR$. DFT calculations also show that the dependence of $\Delta G(*H)$ on the second M is more sensitive than that of $\Delta G(*HOCO)$. Although this study was focused on the case of bimetallic Pd hydrides, the trends from the current study should be also helpful in optimizing other catalytic systems with their different binding affinities toward competing HER and $CO_2RR$ reactions. Furthermore, the promising results on PdNi and PdCu reveal the possibility of reducing Pd loading by alloying Pd with non-precious metals.

## Methods

**Preparation of Pd-based bimetallic nanoparticles on carbon.** All reagents were used without purification. Unless otherwise noted, all of the chemicals were purchased from Sigma Aldrich. The Pd-based bimetallic nanoparticles (NPs) supported on porous carbon (C) were prepared with a simple co-precipitation method[17,49]. First, hexadecyltrimethylammonium bromide (CTAB) functionalized C powder (Vulkan XC-72, Cabot) was prepared by mixing 135 mg of C dispersed in ethylene glycol (EG) with 500 mg of CTAB. The mixture was sonicated for 30 min while the temperature was carefully monitored under 30 °C to prevent an undesired precipitation. Next, 10 wt% Pd-based NPs on C ($Pd_{0.8}M_{0.2}$ on C, M = Co, Ni, Cu, Ag, and Pt) were prepared by adding an aqueous solution (5 mL) containing the calculated amount of the precursors (potassium palladium chloride ($K_2PdCl_4$), cobalt sulfate heptahydrate ($CoSO_4·7H_2O$), nickel sulfate hexahydrate ($NiSO_4·6H_2O$), copper nitrate trihydrate ($Cu(NO_3)_2·3H_2O$), silver nitrate ($AgNO_3$), and tetraammineplatinum nitrate ($Pt(NH_3)_4(NO_3)_2$)) and 1 M sodium hydroxide solution (NaOH, 1 mL). This solution was refluxed at 90 °C for 2 h with vigorous stirring to reduce the precursors to the corresponding metallic NPs. Once the flask was cooled down, the precipitate was filtered and washed by using the vacuum filtration method with ethanol and deionized water. The obtained powder was vacuum-dried overnight at 80 °C. Then, the final product of 10 wt% Pd-based bimetallic NPs on C was obtained and denoted as PdM for the bimetallic cases. Inductively-coupled plasma-optical emission spectroscopy (ICP-OES, Optima 8300, PerkinElmer) confirmed the successful synthesis of bimetallic PdM with desired Pd/M atomic ratios (See Supplementary Table 1). For the Au/C sample, gold chloride ($AuCl_3$) was used as a metal precursor. Except for the reflux temperature (25 °C) and the molarity of NaOH solution (0.01 M), the other conditions are the same with the aforementioned procedure. High resolution transmission electron microscopy (Talos ×200, FEI) was used to characterize the morphologies and elemental distributions of the samples.

**Preparation of working electrode.** For the preparation of working electrode, the obtained powder was dispersed in the mixture of DI water and isopropanol solution

($v:v = 1:1$) containing 0.05 % Nafion with a concentration of 2 mg mL$^{-1}$. After sociation for 20 min, the catalyst ink was dropped on Toray carbon paper (TGP-H-060, 10% waterproofed) and dried. The areal mass loading was 100 μg cm$^{-2}$.

**Electrochemical measurements.** Leak-free Ag/AgCl (EDAQ, ET-072) and graphite paper were used as reference and counter electrodes, respectively. The 0.25 M sodium carbonate ($Na_2CO_3$) solution was bubbled with $CO_2$ gas overnight to prepare 0.5 M sodium bicarbonate ($NaHCO_3$) solution, which was utilized as an electrolyte. The potassium-containing electrolyte (0.5 M $KHCO_3$) was prepared using the same method. The pH values of these electrolytes were 7.35 after saturation. More details regarding the experimental procedures[2,17] are provided in the Supplementary Information.

For obtaining electrochemical surface area (ECSA)[50], CO-stripping experiment was conducted. Each electrode was first cycled in an Ar-saturated 0.1 M NaOH solution for 5 cycles in the range from 0.1 $V_{RHE}$ to 1.1 $V_{RHE}$ with a scan rate of 20 mV s$^{-1}$. Then, CO was adsorbed on the electrode by holding a potential of 0.1 $V_{RHE}$ for 10 min in a CO-saturated 0.1 M NaOH solution. Then, the CO stripping curve was achieved after purging Ar for 20 min in a range from 0.1 $V_{RHE}$ to 1.1 $V_{RHE}$ with a scan rate of 20 mV s$^{-1}$. The charge densities for CO stripping were assumed to be 420 μC cm$^{-2}$. For comparison, the values of ECSA using the reduction capacitance (430 μC cm$^{-2}$) of surface $Pd(OH)_2$ were calculated based on a previous report[51]. The calculated ECSA values are tabulated in Supplementary Table 14.

**In situ measurements.** The lab-made acryl kit was used for the in situ X-ray measurements (Supplementary Fig. 25). The potential range used for the in situ X-ray measurements was determined after confirming the potential range sufficient for transforming Pd into the PdH phase.

In situ XAFS measurements[52,53] were conducted in the 2–2 beamline (for Pd K-edge, Ag K-edge, and Pt L$_3$-edge) at Stanford Synchrotron Radiation Laboratory (SSRL) and in the 9-BM beamline (for Cu K-edge) at Advanced Photon Source (APS). While transmission mode was used at Pd K-edge, fluorescent mode was used at the other element's edges, mainly because of the diluted amount of elemental content. The typical duration for a single spectrum was ~17 min. During all of the XAFS measurements, the spectrum of each reference metal foil (i.e., Pd, Ag, Pt, and Cu) was simultaneously recorded, and was further used for calibrating the edge energy ($E_0$) of the sample under analysis. More details regarding the experimental procedures and data analyses[54] are provided in the Supplementary Information.

In situ XRD measurements were performed in the 17-BM-B beamline at APS. A Perkin-Elmer amorphous silicon detector was used under the transmission mode. The wavelength was 0.24141 Å. The duration elapsed for an individual diffraction pattern was controlled to 2 min. The obtained 2D ring patterns were integrated and converted to 1D diffraction patterns by using GSAS-II software[55]. The areal mass loading was c.a. 8.5 mg cm$^{-2}$ to achieve the sufficient diffraction intensity. A laboratory-made H-shaped acryl cell was used for electrochemical operations. During the measurements, $CO_2$ gas was continuously bubbled into the electrolyte. The other conditions were the same as in the electrochemical measurements. The potential was scanned at a scan rate of 0.05 mV s$^{-1}$ under LSV mode in a range from 0.2 to $-0.7$ $V_{RHE}$. So, the potential interval for each XRD pattern was 6 mV. Every fourth XRD pattern (=24 mV) was chosen in Fig. 3.

**Faradaic efficiency and partial current density calculations.** The Faradaic efficiency (FE) for CO (FE(CO)) was calculated by using the following Eqs. (3–6) [2,17]:

$$N_{CO}(\text{in gaseous phase}) = C_{CO}[\%] \times 17(\text{mL})/22.4 \left(\text{L mol}^{-1}\right) \quad (3)$$

$$N_{CO}(\text{dissolved in electrolyte}) = (K_{CO}) \times 50(\text{mL}) \times 1(\text{g/mL})/18 \left(\text{g mol}^{-1}\right) \times C_{CO}[\%] \quad (4)$$

$$N_{CO}(\text{total}) = N_{CO}(\text{in gaseous phase}) + N_{CO}(\text{dissolved in electrolyte}) \quad (5)$$

$$FE(CO)[\%] = \frac{2 \times 96485 \left(\text{Cmol}^{-1}\right) \times N_{CO}(\text{total})}{\int_0^t i dt} \times 100 \quad (6)$$

Where $N_{CO}$ (in gaseous phase) is the amount of CO in the empty head space at the cathodic compartment of the cell. $N_{CO}$ (dissolved in electrolyte) is the amount of CO dissolved in the catholyte estimated using the Henry's law. The constant ($K_{CO}$ = $1.774 \times 10^{-5}$) is the molar solubility of CO in water at CO partial pressure of 1 bar and room temperature. $C_{CO}$ is the concentration (%) of CO in the empty head space, which is determined from GC analysis. For FE(CO) calculations, Faradaic constant (96485 C mol$^{-1}$) and the number of electron (2) for a single CO production were used. $i$ and $t$ are the current (Ampere) and electrolysis time (seconds) measured by the Potentiostat, respectively. The same procedure was used for calculating FE for $H_2$ (FE ($H_2$)) except for its different molar solubility in water (i.e., $K_{H2} = 1.411 \times 10^{-5}$).

The partial current densities of CO (J(CO)) and $H_2$ (J($H_2$)) were calculated based on the following Eqs. (7 and 8) [2,17]:

$$J(CO)\left[mA\,cm^{-2}\right] = FE(CO) \times \frac{\int_0^t i\,dt}{A} \div 100 \qquad (7)$$

$$J(H_2)\left[mA\,cm^{-2}\right] = FE(H_2) \times \frac{\int_0^t i\,dt}{A} \div 100 \qquad (8)$$

A is the geometrical area of the working electrode. Based on these partial current densities, Tafel plots were constructed in Supplementary Fig. 6. The error bars in FE and J calculations were obtained from repeated gas analyses using GC.

**Computational methods.** Spin unrestricted periodic DFT calculations[38,39] were performed at the GGA level within the PAW-PW91 formalism[37,41] using the Vienna Ab Initio Simulation Package (VASP) code[42,43]. A $5 \times 5 \times 1$ Monkhorst-Pack grid[44] for k-points and a plane wave cut-off energy of 400 eV were used for total energy calculations.

The PdH was modeled using the NaCl crystal structure. The Pd-based bimetallic hydrides ((PdM)H; M = Ag, Cu, Ni, Pt and Co) were modeled using the cubic ($Pd_3M)H_4$-$L1_2$ crystal structures. The energetically most stable low index (111) surface of PdH/PdMH was chosen in the DFT calculations to represent the surface of relatively large nanoparticles (5–10 nm) in the experiments. The H-terminated PdH(111) and $Pd_3MH_4$(111) surfaces were modeled using a six bilayers (a bilayer contains a unit of one Pd/M layer and one H layer) $2 \times 2$ surface slabs. A vacuum layer of ~14 Å thick was added in the slab cell along the direction perpendicular to the surface in order to minimize the artificial interactions between the surface and its periodic images. During geometry optimization, atoms in the top three layers were allowed to relax while atoms in the bottom three layers were fixed until the Hellman-Feynman force on each ion was smaller than 0.01 eV Å$^{-1}$. The binding energy (BE) of an adsorbate was calculated as the following Eq. (9):

$$BE(adsorbate) = E(slab + adsorbate) - E(slab) - E(adsorbate) \qquad (9)$$

where E(slab + adsorbate), E(slab) and E(adsorbate) are the total energy of slab with adsorbate, the energy of clean slab and the energy of adsorbate in the gas phase, respectively.

The Gibbs free energy (G) of a species is calculated as the following Eq. (10)[45]:

$$G = E + ZPE - TS \qquad (10)$$

Here, E is the total energy of a species obtained from DFT calculations, ZPE and S are the zero-point energy and entropy of a species, respectively, and $T = 298.15$ K.

The free energy diagram of $CO_2RR$ to CO is calculated by considering the following sequential steps (11–13) [14,56,57]:

$$CO_2 + (H^+ + e^-) + * \rightarrow *HOCO \qquad (11)$$

$$*HOCO + (H^+ + e^-) \rightarrow *CO + H_2O \qquad (12)$$

$$*CO \rightarrow CO + * \qquad (13)$$

The free energy diagram of the HER, which inevitably takes place in aqueous electrolytes, was calculated via the following sequential steps (14, 15) [14]:

$$H^+ + e^- + * \rightarrow *H\,(Volmer\,step) \qquad (14)$$

$$*H + H^+ + e^- \rightarrow H_2 + *\,(Heyrovsky\,step) \qquad (15)$$

The *OH formation energy on the (PdM)H (111) surface was calculated as the following Eq. (16):

$$OH\,formation\,energy = E[OH-(PdM)H(111)] - E[(PdM)H(111)] - 1/2\,E[O_2] \qquad (16)$$

where, E[OH-(PdM)H(111)], E[(PdM)H(111], and E[$O_2$] are total energies of (PdM)H(111) surfaces with OH, clean (PdM)H(111) surface and $O_2$ molecule in gas phase, respectively.

## Data availability
The data that support the findings of this study are available from the corresponding author upon reasonable request.

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

## Acknowledgements

This research was supported by the US Department of Energy, Basic Energy Science, Catalysis Science Program (Grant No. DE-FG02-13ER16381). We acknowledge technical supports with the 2-2 (XAFS) at Stanford Synchrotron Radiation Laboratory (SSRL, Contract No. DE-AC02-76SF00515), 9-BM (XAFS), and 17-BM-B (XRD) beamlines at Advanced Photon Source (APS, Contract No. DE-AC02-06CH11357). In addition, we acknowledge the technical support and computational resources from Center for Functional Nanomaterials at Brookhaven National Laboratory (Contract No. DE-SC0012704). J.H.L. acknowledges the National Research Foundation of Korea (NRF) funded by the Ministry of Education (Grant No. NRF-2017R1A6A3A03004202).

## Author contributions

J.H.L. and S.K. contributed equally to this work; J.H.L. synthesized materials and carried out the characterizations and electrochemical evaluations; S.K. conducted DFT calculations; J.H.L., Z.J., Z.X., S.Y. and B.M.T. analyzed the gas quantifications; J.H.L., Z.X., S.Y., W.X. and N.S.M. conducted the synchrotron-based experiment; J.H.L., S.K., Z.J. and J.G.C. designed the experiments and analyzed the data; J.H.L., S.K., and J.G.C. wrote the paper; and J.G.C. supervised the whole project.

## Additional information

**Competing interests:** The authors declare no competing interests.

