## [Peer Review File · Nature Communications]

Reviewers' comments:

Reviewer #1 (Remarks to the Author):

In this work, the authors explored electrochemical carbon dioxide reduction reaction on Pd-based bimetallic hydrides. Theoretical and experimental results showed that the second metal had a significant effect on the competitive adsorption of $*H$ and $*HOCO$, leading to controllable syngas ratios. The authors also proposed a descriptor (i.e., $\Delta G(*H) - \Delta G(*HOCO)$) to match the rule with CO₂RR activity and selectivity. However, I am reluctant to recommend its publication on Nature Communications.

First of all, it is well known that syngas can be produced from natural gas, coal, biomass, or virtually any hydrocarbon feedstock. Steam reforming, which was known as an endothermic reaction with 206 kJ/mol energy needed for conversion, was the industrial process for the syngas products. The commercial catalysts used in steam reforming are commonly noble-metal-free catalysts, for example, Fe-based catalysts. As noble metal materials, Pd-based catalysts proposed by the authors exhibited a current density for CO less than 2 mA cm⁻². Production scale and catalyst cost both limit industrial applications. From the perspective of the products, value-added multi-carbon products such as C₂H₄, CH₃COOH, CH₃CH₂OH, etc. is much more attractive.

At the meantime, the mechanism proposed by the author lacked novelty. A similar descriptor has been reviewed (10.1002/cphc.201700736). The descriptor for $*HOCO$ was not a universal descriptor for CO₂ electrochemical reduction, because CO₂ reduction can be achieved not only by $HOCO^*$ but also by $HCOO^*$ (formate mechanism) or direct deoxygenation to produce CO (RWGS). The adsorption energy of O may be fitted for the descriptor for CO₂ electrochemical reduction (10.1038/NCHEM.1873).

Recent review used O affinity and H affinity as the descriptor (10.1016/j.chempr.2018.05.001). Furthermore, I wonder if $\Delta G(*H) - \Delta G(*HOCO)$ could work as an effective descriptor to evaluate the selectivity of CO. Even if the value of $\Delta G(*H) - \Delta G(*HOCO)$ was the same, different selectivity might be obtained. The selectivity of CO₂RR was also related to the value of $\Delta G(*H)$ and $\Delta G(*CO)$ (10.1002/cphc.201700736). The authors need to exclude this kind of possibility.

More importantly, the authors declared that CO₂RR was limited by mass transportation. Hence, the adsorption energy of intermediate species may not virtually reflect energy barrier of the catalytic reaction. The essence of this work should be the exploration of CO₂ diffusion and H adsorption, not $*HOCO$ adsorption.

Specific issues were listed below:

-Although the characterization was very advanced, they reflect the structural information of the catalyst rather than information on the adsorbed species, which was not correlated with the mechanism raised by the authors.

-Morphology characterization was too scarce. HRTEM and mapping image are needed.

-The authors mentioned that the combination of second metal with Pd could affect the formation potential of PdH. However, the role of PdH was not explored. Did PdH participate in CO₂ electrochemical reduction? What role did PdH play during CO₂ electrochemical reduction process?

-Pd was widely studied as a common catalyst towards CO₂ electrochemical reduction into CO. However, I noticed that CO₂ intermediates and H^{*} only bonded with the second metal sites instead of Pd in computational simulation process, so I am wondering the reason.

-The durability of catalysts needed to be discussed. Stability tests of structural and catalytic performance are needed.

Reviewer #2 (Remarks to the Author):

Lee et al. report about the production of syngas from electrochemical CO₂ reduction on Pd –based bimetallic hydrides. Interestingly, a correlation is found between the tendency of the bimetals to form the hydride phase and the CO/H₂ product ratio. DFT calculations identify the difference in Gibbs free energy of H* and *HOCO as the descriptor behind the activity and selectivity towards CO.

The generation of syngas with controlled CO/H₂ ratios from electrochemical CO₂ reduction has been already shown in J. Am. Chem. Soc. 2017, 139, 9359. Here, Cu/Au catalysts with various composition were used and the activity descriptors were identified to be the binding energy of H* and CO*. The authors should at least cite this previous work

The conclusions by Lee et al are potentially interesting, however much more work is needed to make the manuscript acceptable for Nature Communications or any other high impact journal. The following points should be addressed :

- Materials characterization should be performed.

The authors state that they are making alloys with Pd:M=8 :2. Actually there is no evidence that this is the case. For the Vegard's law a peak shift should be observed in the XRD pattern, which does not seem to be the case. The composition should be confirmed by elemental analysis (ICP-OES or MS). STEM-EDX mapping should be performed before and after catalysis to assure that there is a uniform distribution of the two metals within the nanoparticles.

- Studies on electrochemical CO₂ reduction are booming and it is important that we are able to compare results across the literature if we really want to assure progress in the field. That said, potassium bicarbonate is the most common electrolyte in the literature so far, yet sodium bicarbonate is used here. The authors should demonstrate that the behaviour of their catalysts is preserved independently of the electrolyte used.

- The analysis of the liquid products is needed when the FE % is lower than 100% to assure that no other redox process is occurring.

- It is concerning that the authors are entering into a regime controlled by mass transport of CO₂ and it is difficult to say that -0.9V vs RHE is the last potential before that happens. This is quite of a problem if we are discussing the intrinsic activity of the catalysts. Can the authors repeat the measurements while stirring the electrolyte ? Later on in Figure 4F J(CO) @-1.0 V vs RHE is reported, which is not acceptable if indeed we are in a mass transport limited regime.

- Again for the sake of comparison across the literature, the designs of the electrochemical cell and of the in-situ cells should be added in the SI. Furthermore, it should be demonstrated that the conditions in the in-situ cells are the same of the electrochemical cells so to actually lead to the same product distribution.

- It is noted that the potential reached in the in-situ experiments is not the same of the electrochemical measurements. Why is this the case ?

- The DFT calculations were performed only on (111) surfaces. The authors write « The particle size

for all samples is in the range from 5 to 10 nm (Supplementary Fig. 13), thus enabling us to use their (111) surface as a platform for further calculations." It is not clear why this is the case. In fact, spherical nanoparticles are most likely to expose all the facets on their surface. See for example JACS 2014, 136, 6978. Can the authors truly justify their statement (perhaps with high res TEM) or otherwise repeat their calculations also on (100) and (110)?

- Can the theory or the experiments illustrate how the trends will change with composition within the same alloy?
- How does the ECSA from CO stripping compared with the ECSA from double-layer capacitance measurements which is usually employed for Cu-based electrocatalysts? The actual ECSA values should be reported in a table.
- Finally the standard deviation should be calculated on multiple samples not on multiple GC measurement on the same sample. Values should be corrected.

Reviewer #3 (Remarks to the Author):

In this paper, electrocatalytic CO₂RR and HER using M doped PdH catalysts (M = Co, Cu, Ni, Ag, Pd, and Pt) were studied both experimentally and computationally. Especially, the discussion regarding the correlation between experimental J(CO) and computational $\Delta G(*H) - \Delta G(*HOCO)$ values is interesting. However, there still are some points that need to be clarified as listed below. Once these points are addressed I can recommend publication of this paper in Nature Communications.

1. In this study, H* and HOCO* were assumed to adsorb on M rather than Pd. Depending on M, the adsorption site should change. I therefore recommend to discuss $\Delta G(*H/*HOCO)$ for adsorption on Pd sites.

2. The statistical analysis needs to be performed. The authors stated that there is a good correlation between J(CO) and $\Delta G(*H) - \Delta G(*HOCO)$ and $\Delta G(*H) - \Delta G(*HOCO)$ can be a good descriptor for this reaction. However, there is no quantitative evidence showing that there is a good correlation between J(CO) and $\Delta G(*H) - \Delta G(*HOCO)$. I see some correlation also between J(CO) and $\Delta G(*HOCO)$. I also see some correlation between J(CO) and $\Delta G(*H) - \Delta G(*CO)$. How did the authors judge that there is a good correlation or not? Just from their impression? Only from these figures, it is not clear how the authors found correlations among various combinations of experimental and computational values. I recommend to analyze correlations more quantitatively

3. In this study, only six M were considered. Among the six, Pt doesn't show a good correlation. I suppose that there should be many other cases where the $\Delta G(*H) - \Delta G(*HOCO)$ value is not a good descriptor. I suggest the authors to discuss applicability of the descriptor.

Overall Response:

We thank all the reviewers for their valuable comments. As described below, we have performed additional experiments and DFT calculations to address all the comments. We would like to clarify that, because our manuscript includes six Pd-M bimetallic systems, it was impossible to perform additional studies on all catalysts with different Pd/M ratios or in different electrolytes. Instead we selected a representative bimetallic system, Pd-Ni, to perform all the additional experimental and DFT studies to answer the relevant questions by the reviewers.

Reviewer #1 (Remarks to the Author):

In this work, the authors explored electrochemical carbon dioxide reduction reaction on Pd-based bimetallic hydrides. Theoretical and experimental results showed that the second metal had a significant effect on the competitive adsorption of *H and *HOCO, leading to controllable syngas ratios. The authors also proposed a descriptor (*i.e.*, $\Delta G(*H) - \Delta G(*HOCO)$) to match the rule with CO₂RR activity and selectivity. However, I am reluctant to recommend its publication on Nature Communications.

Response: We thank the reviewer for his/her thorough review of our manuscript and helpful comments.

1. First of all, it is well known that syngas can be produced from natural gas, coal, biomass, or virtually any hydrocarbon feedstock. Steam reforming, which was known as an endothermic reaction with 206 kJ/mol energy needed for conversion, was the industrial process for the syngas products. The commercial catalysts used in steam reforming are commonly noble-metal-free catalysts, for example, Fe-based catalysts. As noble metal materials, Pd-based catalysts proposed by the authors exhibited a current density for CO less than 2 mA cm⁻². Production scale and catalyst cost both limit industrial applications. From the perspective of the products, value-added multi-carbon products such as C₂H₄, CH₃COOH, CH₃CH₂OH, *etc.* is much more attractive.

Response: The novelty in this work lies in the finding that the syngas ratio (*i.e.*, selectivity) and the current density (*i.e.*, yield) toward each product can be controlled at the same potentials even by reducing the amount of Pd. At the same time, we were able to propose a potential descriptor (*i.e.*, $\Delta G(*H) - \Delta G(*HOCO)$) for explaining both CO₂RR and HER. A current density for CO ($J(\text{CO})$) less than 2 mA cm⁻², which would be the reviewer's major concern, can be indeed enhanced by the optimization process such as the catalyst loading, the cell design, and the choice of electrolyte. For example, as an effort to increase the $J(\text{CO})$, we tested Pd and PdNi at -0.9 V_{RHE} using 0.5M KHCO₃, leading to 4.9~6.8 times higher $J(\text{CO})$ (4.46 and 4.68 mA cm⁻², respectively) when compared with using 0.5M NaHCO₃. Moreover, the FE(CO) values (76.7 and 85.1 %, respectively) were enhanced by ~1.5 times. This is because of the difference in hydrolysis capability of the hydrated cation, which is well explained in the literature (**Ref. 21-22** in the revised manuscript). Regarding the multiple hydrocarbon production, it is rarely feasible using Pd-based catalysts. Yet, in considering the potential cost for the separation process originating from the low selectivities toward various hydrocarbon products (frequently, observed in Cu-based catalysts in **Ref. 21-24**), the

electrochemical syngas production over bimetallic Pd hydrides might be a potentially more viable option because the produced syngas can be used directly as the feed for subsequent methanol synthesis or Fischer-Tropsch processes.

Action: In response to the reviewer's comment, we revised our manuscript as below:

[Page 6]

The FE(CO) and CO/H₂ ratio follow the trend...compared with the others. This trend remains the same even with a different electrolyte (*i.e.*, 0.5M KHCO₃) while the values of J(CO) and FE(CO) are enhanced significantly due to the different hydrolysis effects²¹⁻²² of Na⁺ and K⁺ (Supplementary Fig. 8). For comparison, 10wt% Au NPs...

[Experimental Section on page 15]

The 0.25M sodium carbonate (Na₂CO₃) solution was bubbled with CO₂ gas overnight... as an electrolyte. The potassium-containing electrolyte (0.5M KHCO₃) was prepared using the same method. The pH values of these electrolytes were 7.35 after saturation.

[Supplementary Information]

Supplementary Figure 8. The FE(CO) and J(CO) for (A) Pd and (B) PdNi at -0.9 V_{RHE} in CO₂-saturated 0.5M NaHCO₃ and KHCO₃ electrolytes.

2. At the meantime, the mechanism proposed by the author lacked novelty. A similar descriptor has been reviewed (10.1002/cphc.201700736). The descriptor for *HOCO was not a universal descriptor for CO₂ electrochemical reduction, because CO₂ reduction can be achieved not only by HOCO* but also by HCOO* (formate mechanism) or direct deoxygenation to produce CO (RWGS). The adsorption energy of O may be fitted for the descriptor for CO₂ electrochemical reduction (10.1038/NCHEM.1873). Recent review used O affinity and H affinity as the descriptor (10.1016/j.chempr.2018.05.001). Furthermore, I wonder if $\Delta G(*H) - \Delta G(*HOCO)$ could work as an effective descriptor to evaluate the selectivity of CO. Even if the value of $\Delta G(*H) - \Delta G(*HOCO)$ was the same, different selectivity might be obtained. The selectivity of CO₂RR was also related to the value of $\Delta G(*H)$ and $\Delta G(*CO)$ (10.1002/cphc.201700736). The authors need to exclude this kind of possibility.

Response: We thank the reviewer for the insightful comments about the potential descriptors

of CO₂ conversion. The CO₂RR may occur *via* various pathways to produce a wide range of products. It has been shown that the CO production primarily occurs *via* the carboxylic *HOCO intermediate (Sun *et al.*, *J. Am. Chem. Soc.* 2013, 135, 16833) while the formation of formate (*HCOO) intermediate leads to the formation of formic acid as a final product (Sargent *et al.*, *Joule*, Volume 1, Issue 4, 20 December 2017, Pages 794-805). Therefore, the DFT calculations in the present study were carried out to study CO₂ conversion to CO *via* the *HOCO intermediate. Along this reaction pathway, *HOCO and *CO are two key intermediates for the formation of CO. This provides a natural choice of using the binding energy of *HOCO and/or *CO as a potential descriptor of CO₂RR. However, our calculated binding energies of *HOCO and *CO do not correlate well with the experimental selectivity among various (PdM)H catalysts. Thus, we conclude that *HOCO and *CO binding energies alone may not serve as descriptors of CO₂RR on the (PdM)H catalysts. In contrast, we find that the binding energy difference between *H (key intermediate in HER) and *HOCO (key intermediate in CO₂RR), or the $\Delta G(*H) - \Delta G(*HOCO)$, correlates well with the experimentally observed selectivity. Thus, we propose $BE(*H) - BE(*HOCO)$ or $\Delta G(*H) - \Delta G(*HOCO)$ as a potential descriptor of selectivity for the (PdM)H catalysts.

Following the reviewer's suggestion, we performed additional DFT calculations to determine the O binding energy on the (PdM)H surfaces. Our calculations show that O is not stable on the (PdM)H(111) surfaces. The surface O reacts with nearby surface H to form *OH. Therefore, we calculated *OH formation energies on (PdM)H(111) surfaces to test the possibility of using *OH formation energy as a potential descriptor. The *OH formation energy is calculated as:

$$\text{OH formation energy} = E[\text{OH-(PdM)H(111)}] - E[(\text{PdM)H(111)}] - \frac{1}{2} E[\text{O}_2]$$

where, $E[\text{OH-(PdM)H(111)}]$, $E[(\text{PdM)H(111)}]$ and $E[\text{O}_2]$ are total energies of (PdM)H(111) surfaces with OH, clean (PdM)H(111) surface and O₂ molecule in gas phase, respectively.

The DFT calculated OH formation energies (**Supplementary Table 14**) do not correlate well with the experimental results (**Supplementary Fig. 24**). Therefore, we exclude the possibility of using O/OH binding energy as potential descriptors in our study.

Action: Following the reviewer's suggestion, additional DFT calculations were performed to test the possibility of using the *OH formation energy as a potential descriptor. We have revised the manuscript and supplementary information as below:

[Page 12]

... However, the same consideration using *H or *CO adsorption alone does not correlate with $J_{\text{ECSA}}(\text{CO})$ or $J_{\text{ECSA}}(\text{H}_2)$ as displayed in Supplementary Fig. 22 and Supplementary Fig. 23. The binding energy of O has been predicted as a potential descriptor of CO₂RR.⁴⁶⁻⁴⁷ However, in the current study O reacts with surface H to form OH on (PdM)H. The DFT calculated OH formation energy does not correlate with the experimental $J_{\text{ECSA}}(\text{CO})$ or $J_{\text{ECSA}}(\text{H}_2)$ (Supplementary Fig. S24 and Table 14), suggesting that the binding energy of O/OH is not a descriptor for CO₂RR over the (PdM)H catalysts.

[Computational Methods on page 19]

The *OH formation energy on the (PdM)H (111) surface was calculated as:

$$\text{OH formation energy} = E[\text{OH-(PdM)H(111)}] - E[(\text{PdM})\text{H(111)}] - \frac{1}{2} E[\text{O}_2]$$

where, $E[\text{OH-(PdM)H(111)}]$, $E[(\text{PdM})\text{H(111)}]$ and $E[\text{O}_2]$ are total energies of $(\text{PdM})\text{H(111)}$ surfaces with OH, clean $(\text{PdM})\text{H(111)}$ surface and O_2 molecule in gas phase, respectively.

[Supplementary Information]

Supplementary Figure 24. Correlations between OH formation energies and (A) $J_{\text{ECSA}}(\text{CO})$ and (B) $J_{\text{ECSA}}(\text{H}_2)$ at $-0.9 \text{ V}_{\text{RHE}}$. The same correlation constructed by using (C) $J_{\text{ECSA}}(\text{CO})$ and (B) $J_{\text{ECSA}}(\text{H}_2)$ at $-1.0 \text{ V}_{\text{RHE}}$.

[Supplementary Information]

Supplementary Table 14. DFT calculated OH formation energies.

Entry	Formation energies (eV)
(PdAg)H	-1.94
(PdCu)H	-2.24
(PdNi)H	-2.02
PdH	-1.16
(PdCo)H	-1.97
(PdPt)H	-1.64

3. More importantly, the authors declared that CO_2RR was limited by mass transportation. Hence, the adsorption energy of intermediate species may not virtually reflect energy barrier of the catalytic reaction. The essence of this work should be the exploration of CO_2 diffusion and $^*\text{H}$ adsorption, not $^*\text{HOCO}$ adsorption.

Response: We thank the reviewer for the careful comments. As the reviewer pointed out,

CO₂RR is generally controlled by the mass transport of CO₂ because of the limited amount of CO₂ in aqueous electrolytes. In an attempt to avoid this issue, the vigorous stirring was applied during CO₂RR. Therefore, the CO₂ diffusion in this study, if any, should not vary with different catalysts because all of the samples were measured under the same conditions.

We also agree that *H adsorption should be important in this study because of its relative abundance during electrolysis. This is also consistent with the linearity in J_{ECSA}(H₂) vs. ΔG(*H) consideration. However, the trend observed for J_{ECSA}(CO) cannot be explained by any individual intermediate (See **Supplementary Fig. 20 and 21**). For this reason, we introduced the combined effect of *H and *HOCO adsorption to explain both CO₂RR and HER. Even though this consideration may not quantitatively reflect the actual energy barrier of the given electrocatalytic reactions, it is the most reasonable descriptor because only this consideration can explain both the trends in CO₂RR and HER.

Action: The absence of a linear correlation between J(CO) and ΔG values of individual intermediates is demonstrated in the revised manuscript and SI. The relevant sentences were added in the revised manuscript as described below:

[Page 5]

The CO₂RR activity...0.5M sodium bicarbonate (NaHCO₃) electrolyte with vigorous magnetic stirring at different potentials (Supplementary Fig. 3). The gaseous product...

[Page 11]

As an effort to find a key descriptor for the CO₂RR with controlled CO/H₂ ratios,...in Supplementary Fig. 20 and Fig. 21 (See the Methods and Supplementary Table 13). It was noted that, while ΔG(*H) could be linearly correlated with J_{ECSA}(H₂), J_{ECSA}(CO) could not be scaled with any of the individual value of ΔG (*H), ΔG(*HOCO) or ΔG(*CO).

[Supplementary Methods in Supplementary Information]

After the additional CO₂ bubbling for 10 min, the electrochemical CO₂RR performance was evaluated...for a designated duration. The vigorous magnetic stirring was applied during the electrolysis to help mitigate the mass transport limitation of dissolved CO₂ in the electrolyte. With an increase...

[Supplementary Information]

Supplementary Figure 21. Correlations between $J_{\text{ECSA}}(\text{CO})$ at $-0.9 \text{ V}_{\text{RHE}}$ and free energies of (A) $*\text{H}$, (B) $*\text{HOCO}$, and (C) $*\text{CO}$. The same correlation constructed by using $J_{\text{ECSA}}(\text{CO})$ at $-1.0 \text{ V}_{\text{RHE}}$.

4. Specific issues were listed below:

4.1 Although the characterization was very advanced, they reflect the structural information of the catalyst rather than information on the adsorbed species, which was not correlated with the mechanism raised by the authors.

Response: The electroreduction mechanism of CO_2 -to- CO has been relatively well established in the literature. Typically, CO is formed through a carboxylic intermediate ($*\text{HOCO}$ in this study) with the overall pathway as $*+\text{CO}_2 \rightarrow *\text{HOCO} \rightarrow *\text{CO} \rightarrow *+\text{CO}$ (Ref. 14, 56-57). However, in considering that the Pd surface is vulnerable to CO -poisoning, it is unlikely that Pd could produce CO because CO desorption would be a rate-limiting step. Nonetheless, it is possible to produce CO on Pd-based catalysts because Pd undergoes the phase transition to form PdH under the CO_2RR condition and consequently its binding affinity of CO is reduced. We revealed that all bimetallic Pd-M (except for $\text{M}=\text{Pt}$) can be transformed to bimetallic PdH by using in-situ X-ray analyses, leading to free energy calculations on the PdH phase. Therefore, we believe that our experimental characterization is well-consistent with the mechanisms proposed from DFT calculations.

Action: To address this issue, we added the following sentence in the main text.

[Page 5]

... CO and H_2 were the two major products. It is well known that Pd-based catalysts are vulnerable to CO -poisoning^{2,32-33} due to its strong binding to CO . However, the transformation of Pd-to-PdH under the CO_2RR condition, which will be discussed later, reduces the binding energy of CO and thus enables facile CO desorption. The Faradaic efficiencies (FEs) of...

4.2 Morphology characterization was too scarce. HRTEM and mapping image are needed.

Response: Following the reviewer's suggestion we performed additional TEM analyses for Pd and PdNi before and after CO₂RR. As displayed below (**Supplementary Fig. 9** in the revised version), both samples showed the spherical morphology and maintained similar particle size distribution after CO₂RR. The element mapping was also conducted for PdNi, indicating that Pd and Ni were in close proximity, consistent with the formation of PdNi alloy both before and after CO₂RR (**Supplementary Fig. 10**).

Action: Following the reviewer's suggestion, STEM and elemental mapping results are included in the revised Supplementary Information. The following sentences are added in the revised manuscript.

[Page 6]

...representing a better catalyst for syngas production than Au/C. The stability of the Pd and PdNi catalysts was characterized using Transmission electron microscopy (Supplementary Fig. 9), which revealed that both Pd and PdNi maintained the original particle size after CO₂RR. Furthermore, the elemental mapping verified that Pd and Ni were in close proximity, consistent with the formation of PdNi alloy both before and after CO₂RR (Supplementary Fig. 10).

[Page 14]

... are the same with the aforementioned procedure. High resolution transmission electron microscopy (Talos 200x, FEI) was used to characterize the morphologies and elemental distributions of the samples.

[Supplementary Information]

Supplementary Figure 9. (A) Partial current density ($J(\text{CO})$) and Faradaic efficiency ($\text{FE}(\text{CO})$) of CO during CO₂RR at $-0.9 \text{ V}_{\text{RHE}}$ with Pd for 2 h. STEM images of a Pd catalyst taken (B) before and (C) after CO₂RR. (D-F) The same analyses of a PdNi catalyst.

[Supplementary Information]

Supplementary Figure 10. STEM-EDX mapping images of Pd (red) and Ni (green) before and after CO₂RR.

4.3 The authors mentioned that the combination of second metal with Pd could affect the formation potential of PdH. However, the role of PdH was not explored. Did PdH participate in CO₂ electrochemical reduction? What role did PdH play during CO₂ electrochemical reduction process?

Response: This comment is in the same context as Comment 4.1. Bimetallic Pd hydride, (PdM)H is concluded as the active phase for CO₂RR. Based on the correlation of CO production and *in-situ* XRD measurements, CO is not a major product without the formation of the PdH phase. Moreover, the *in-situ* XRD results also reveal that the formation potential of PdH can be tuned by the choice of the second metal, which in turn tunes the syngas ratio between CO and H₂. Therefore, the bimetallic PdH formation plays an important role in the syngas production with controllable ratios.

Action: We added the following sentences in the revised manuscript.

[Page 8-9]

...On the other hand, *in-situ* XRD analysis for PdPt shown...in preventing the hydride formation.³⁴ Therefore, the *in-situ* XRD results reveal that the formation of PdH can be tuned by the choice of the second M. Moreover, the tendency of (PdM)H formation potentials on the choice of the second M coincides with their syngas ratios between CO and H₂ (Figure 1B). Therefore, the bimetallic PdH formation plays an important role in the syngas production with controllable CO/H₂ ratios.

4.4 Pd was widely studied as a common catalyst towards CO₂ electrochemical reduction into CO. However, I noticed that CO₂ intermediates and H* only bonded with the second metal sites instead of Pd in computational simulation process, so I am wondering the reason.

Response: The DFT calculations were performed to calculate binding energies of *H, *HOCO and *CO on all possible adsorption sites on (PdM)H(111) surfaces. **Supplementary Table 12** in the revised Supplementary Information summarizes the DFT calculated binding energies on various adsorption sites. Interestingly, we find that the difference in binding energies (*i.e.*, BE(*H)–BE(*HOCO)) on the second metal sites correlate well with the experimental results. Therefore, we chose to use binding energies calculated on the second metal sites to correlate the trends in HER and CO₂RR on the bimetallic catalyst surfaces.

Action: In response to the reviewer’s suggestion, we have revised the manuscript as below:

[Page 11]

...thus enabling us to use their (111) surface as a platform for further calculations. The binding energies of intermediates calculated on second M sites are used in the DFT discussion to study the immediate effect on HER and CO₂RR due to the presence of second M on the bimetallic catalyst surfaces (Supplementary Table 12). The DFT calculations reveal a correlation between...

[Supplementary Information]

Supplementary Table 12. DFT calculated binding energies (in eV) of *H, *CO and *HOCO on various binding sites.

Intermediates	*H			*CO			*HOCO		BE(*H) - BE(*HOCO) (M sites) ^{a)}
	top-Pd	top-M	hollow	top-Pd	top-M	hollow	top-Pd	top-M	
(PdAg)H	0.27	1.43	0.19	-0.27	-0.10	-0.19	-1.36	-0.88	2.31
(PdCu)H	0.50	1.19	0.38	-0.35	-0.38	-0.03	-1.43	-1.07	2.26
(PdNi)H	0.61	0.48	0.25	-0.33	-0.77	-0.03	-1.41	-1.55	2.03
PdH	0.59	-	0.42	-0.27	-	-0.18	-1.37	-	1.96
(PdCo)H	0.58	-0.03	0.17	-0.36	-1.26	-0.44	-1.40	-1.74	1.71
(PdPt)H	0.54	0.13	0.16	-0.55	-0.44	-0.71	-1.40	-1.71	1.84

^{a)} The BE(*H) at the top sites was used to calculate BE(*H)-BE(*HOCO).

4.5 The durability of catalysts needed to be discussed. Stability tests of structural and catalytic performance are needed.

Response: This comment is in the same context as Comment 4.2. Following the reviewer’s suggestions, we performed short-term CO₂RR stability tests (~2 h) for Pd and PdNi. As displayed in **Supplementary Fig. 9**, both samples clearly showed the stable CO₂RR performance in terms of J(CO) and FE(CO). Our batch-type electrochemical cell did not allow us to perform tests longer than 2 h because the dissolved CO₂ became under-saturated over prolonged cycles. (Marshall *et al.*, *Electrochem. Commun.*, 2019, 101, 78-81) We will design flow-type configurations in our future studies. In addition, as described in the response in Comment 4.2, we performed additional STEM measurements (**Supplementary Fig. 10**)

after CO₂RR to confirm that the distribution of Pd and PdNi NPs did not change.

Action: Please see Response and Action for Comment 4.2.

Reviewer #2 (Remarks to the Author):

Lee et al. report about the production of syngas from electrochemical CO₂ reduction on Pd-based bimetallic hydrides. Interestingly, a correlation is found between the tendency of the bimetallics to form the hydride phase and the CO/H₂ product ratio. DFT calculations identify the difference in Gibbs free energy of H* and *HOCO as the descriptor behind the activity and selectivity towards CO.

Response: We thank the reviewer for the constructive review of our manuscript and helpful comments.

1. The generation of syngas with controlled CO/H₂ ratios from electrochemical CO₂ reduction has been already shown in *J. Am. Chem. Soc.* 2017, 139, 9359. Here, Cu/Au catalysts with various composition were used and the activity descriptors were identified to be the binding energy of H* and CO*. The authors should at least cite this previous work.

Response: We thank the reviewer for bringing up the referred paper to our attention. This JACS paper (Ref. 48 in the revised manuscript) reported a useful strategy toward controlled CO/H₂ ratios using Cu underpotential deposition (UPD) on Au foil as a function of Cu coverage. Ross *et al.* suggested the difference in Gibbs free energy of *H and *CO adsorptions ($\Delta G(*H) - \Delta G(*CO)$) as a potential descriptor for controllable syngas production. However, the calculated $\Delta G(*H) - \Delta G(*CO)$ could not describe the trend of the obtained J(CO) using Cu UPD on Au nanoneedles while it could explain the competition between HER and CO₂RR. This is evidenced by the fact that those samples with similar $\Delta G(*H) - \Delta G(*CO)$ values (*i.e.*, Au $\Theta_{Cu} \sim 1/3$ and Au $\Theta_{Cu} \sim 3/3$, see Figure below) showed different trends on the CO/H₂ ratio (See the Figure below).

[Redacted]

Figure. (left) The obtained J(CO) and J(H₂) and (right) the calculated $\Delta G(*H)$ and $\Delta G(*CO)$ with different amount of Cu UPD on Au. All of the figures reproduced from JACS, 2017, 139, 9359.

Action: To reflect the above discussion, we cited this paper and we revised as below:

[Page 12]

...However, the same consideration using *H or *CO adsorption alone does not correlate with J_{ECSA}(CO) or J_{ECSA}(H₂) as displayed in Supplementary Fig. 22 and Supplementary Fig. 23. The binding energy of O has been predicted... suggesting that the binding energy of O/OH is not a descriptor for CO₂RR over the (PdM)H catalysts. Similarly, a previous study⁴⁸ did not observe a consistent trend between $\Delta G(*H) - \Delta G(*CO)$ and J(CO)/J(H₂) of Cu-

deposited Au. Thus, the combined experimental observations and DFT calculations...

2. The conclusions by Lee et al are potentially interesting, however much more work is needed to make the manuscript acceptable for Nature Communications or any other high impact journal. The following points should be addressed:

2.1. Materials characterization should be performed.

The authors state that they are making alloys with Pd:M=8 :2. Actually there is no evidence that this is the case. For the Vegard's law a peak shift should be observed in the XRD pattern, which does not seem to be the case. The composition should be confirmed by elemental analysis (ICP-OES or MS). STEM-EDX mapping should be performed before and after catalysis to assure that there is a uniform distribution of the two metals within the nanoparticles.

Response: In order to visualize the peak shift as a result of the incorporation of the second M, we compared the peak location corresponding to the (111) and (200) planes (See the revised **Supplementary Fig. 2**). As shown in this Figure, in the cases of using M with larger atomic radius than Pd (M=Ag and Pt), those peaks shift toward lower 2θ values. On the other hand, those peaks shift toward higher 2θ values in the cases of using M with smaller atomic radius (M=Cu, Ni, and Co). This is consistent with the Vegard's law.

As for the elemental analysis, we performed the STEM-EDX mapping for PdNi before and after CO₂RR for 2 h at -0.9 V_{RHE}, as described earlier in the Response for Comment 4.2 by Reviewer 1. Regarding the composition analysis, the STEM-EDX images, EXAFS fitting, and XRD results all suggest the formation of bimetallic alloys.

Action: To reflect the reviewer's question, we have revised the main text and Supplementary Information as below:

[Page 4-5]

The successful synthesis of bimetallic Pd alloys...the same face-centered cubic structure (space group: Fm-3m). The peak locations for the (111) and (200) planes were identified to follow the Vegard's law. In the cases of using M with larger atomic radius than Pd (M=Ag and Pt), those peaks shift toward lower 2θ values. On the other hand, those peaks shift toward higher 2θ values in the cases of using M with smaller atomic radius (M=Cu, Ni, and Co) (Supplementary Fig. 2B).

[Page 6]

...representing a better catalyst for syngas production than Au/C. The stability of the Pd and PdNi catalysts was characterized using Transmission electron microscopy (Supplementary Fig. 9), which revealed that both Pd and PdNi maintained the original particle size after CO₂RR. Furthermore, the elemental mapping verified that Pd and Ni were in close proximity, consistent with the formation of PdNi alloy both before and after CO₂RR (Supplementary Fig. 10).

[Page 14]

... are the same with the aforementioned procedure. High resolution transmission electron microscopy (Talos 200x, FEI) was used to characterize the morphologies and elemental distributions of the samples.

[Supplementary Information]

Supplementary Figure 2. (A) Synchrotron Powder X-ray diffraction (XRD) patterns for all the samples in this study. (B) Magnified XRD patterns in the 2θ range from 5.5 to 7.5°.

[Supplementary Information]

Supplementary Figure 9. (A) Partial current density ($J(\text{CO})$) and Faradaic efficiency (FE(CO)) of CO during CO₂RR at -0.9 V_{RHE} with Pd for 2 h. STEM images of a Pd catalyst taken (B) before and (C) after CO₂RR. (D-F) The same analyses of a PdNi catalyst.

[Supplementary Information]

Supplementary Figure 10. STEM-EDX mapping images of Pd (red) and Ni (green) before and after CO₂RR.

2.2. Studies on electrochemical CO₂ reduction are booming and it is important that we are able to compare results across the literature if we really want to assure progress in the field. That said, potassium bicarbonate is the most common electrolyte in the literature so far, yet sodium bicarbonate is used here. The authors should demonstrate that the behaviour of their catalysts is preserved independently of the electrolyte used.

Response: As the reviewer suggested, we tested and compared Pd and PdNi at -0.9 V_{RHE} with 0.5M NaHCO₃ and 0.5M KHCO₃. As shown below, both samples exhibited enhanced FE(CO) as well as J(CO), which are attributed to the difference in the hydrolysis capability of the hydrated cations (Ref. 21-22). Yet, PdNi still showed better performance than Pd in terms of both FE(CO) (85.1 vs. 76.7%) and J(CO) (4.68 vs. 4.46 mA cm⁻²).

Action: In response to the reviewer's comment, we revised our manuscript as below:

[Page 6]

The FE(CO) and CO/H₂ ratio follow the trend...compared with the others. This trend remains the same even with a different electrolyte (*i.e.*, 0.5M KHCO₃) while the values of J(CO) and FE(CO) are enhanced significantly due to the different hydrolysis effects²¹⁻²² of Na⁺ and K⁺ (Supplementary Fig. 8). For comparison, 10wt% Au NPs...

[Experimental Section on page 15]

The 0.25M sodium carbonate (Na₂CO₃) solution was bubbled with CO₂ gas overnight... as an electrolyte. The potassium-containing electrolyte (0.5M KHCO₃) was prepared using the same method. The pH values of these electrolytes were 7.35 after saturation.

[Supplementary Information]

Supplementary Figure 8. The FE(CO) and J(CO) for (A) Pd and (B) PdNi at $-0.9 V_{RHE}$ in CO_2 -saturated 0.5M $NaHCO_3$ and $KHCO_3$ electrolytes.

2.3. The analysis of the liquid products is needed when the FE % is lower than 100% to assure that no other redox process is occurring.

Response: As the reviewer suggested, we performed liquid product quantification by using 1H NMR analysis. The formic acid ($HCOOH$) was the only product observed in the NMR spectra. The FE of $HCOOH$ accounted for 5~20% from -0.6 to $-0.8 V_{RHE}$ and became negligible at $-0.9V_{RHE}$ and thereafter. This trend is consistent with the results reported elsewhere on Pd-based electrocatalysts (Ref. 2, 30-31). Furthermore, none of the samples exhibited a total FE exceeding 100%.

Action: Following the reviewer's suggestion, we have revised the manuscript as below:

[Page 5]

... CO and H_2 being the major products. Formic acid ($HCOOH$) as a minor product was the only liquid product at all of the potentials, which thus accounted for the rest of FE. Its quantification was done by using 1H NMR measurements (See Supplementary Information). The FE values of $HCOOH$ were determined to be 5~20% from -0.6 to $-0.8 V_{RHE}$ and became negligible at $-0.9 V_{RHE}$ and thereafter. As the potential is ...

[Supplementary Information]

... was collected and injected into GC to quantify the gaseous products. The liquid products were quantified by using 1H NMR spectra with an Avance III spectrometer (Bruker) operating at 400 MHz. Typically, 500 μL of electrolyte taken at the conclusion of the electrolysis was mixed with 10 μL of D_2O and 10 μL of internal standard solution. 2,2,3,3-d(4)-3-(Trimethylsilyl)propionic acid sodium salt (Alfa Aesar) was used as the internal standard (10 mM in D_2O). The 1H NMR spectrum was measured in the water suppression mode. The peak corresponding to $HCOOH$ was detected around a chemical shift of ~ 8.4 ppm. Before switching to...

2.4. It is concerning that the authors are entering into a regime controlled by mass transport of CO_2 and it is difficult to say that $-0.9V$ vs RHE is the last potential before that happens. This

is quite of a problem if we are discussing the intrinsic activity of the catalysts. Can the authors repeat the measurements while stirring the electrolyte? Later on in Figure 4F $J(\text{CO})$ @ -1.0 V vs RHE is reported, which is not acceptable if indeed we are in a mass transport limited regime.

Response: We should clarify that vigorous stirring was indeed applied for all of the samples in the same manner during CO_2RR . In addition, we also confirmed the similar results at $-0.9 V_{\text{RHE}}$ by testing multiple electrodes as shown in the Figure shown below.

Regarding the benchmarking potential(s) where the CO_2 activities are compared, it is somewhat difficult to define it mainly because of (1) the competing natures of CO_2RR (*i.e.*, CO vs. HCOOH vs. H_2 in this study) and (2) the different kinetics for generating different products. This is probably why the CO_2RR community mostly adopts the convention of using the potential where the maximum is achieved to describe catalyst performance. This is supported by the recently reported studies on the combined experimental and theoretical CO_2RR studies; Feaster *et al.* (*ACS Catal.*, 2017, 7, 4822) compared $J(\text{CO}/\text{HCOOH})$ at $-0.9 V_{\text{RHE}}$ whereas Resasco *et al.* (*JACS*, 2017, 139, 11277) and Singh *et al.* (*JACS*, 2016, 138, 13006) compared $J(\text{CO}/\text{H}_2/\text{C}_x\text{H}_y\text{O}_z)$ at $-1.0 V_{\text{RHE}}$. Along with this practice, our comparisons at $-0.9 V_{\text{RHE}}$ (**Supplementary Figure 22**) and $-1.0 V_{\text{RHE}}$ (**Figure 4F** and **Supplementary Figure 23**) seem reasonable because most of the samples in our study generally exhibited the maximum $\text{FE}(\text{CO})$ at these potentials with the sum of $\text{FE}(\text{CO})$ and $\text{FE}(\text{H}_2)$ being close to $\sim 100\%$.

We should admit that our activities cannot be directly compared with the reported values elsewhere because FE and J values might vary with different experimental setups (Clark *et al.*, *ACS Catal.*, 2018, 8, 6560, Ahangari *et al.*, *Electrochem. Commun.*, 2019, 101, 78). However, our electrochemical data, especially the trend among different samples, should be reliable within our own electrochemical setup.

Action: In response to the reviewer's comment, we revised our manuscript as below:

[Page 5]

The CO_2RR activity...0.5M sodium bicarbonate (NaHCO_3) electrolyte with vigorous magnetic stirring at different potentials (Supplementary Fig. 3). The gaseous product...

[Supplementary Methods in Supplementary Information]

After the additional CO_2 bubbling for 10 min, the electrochemical CO_2RR performance was evaluated...for a designated duration. The vigorous magnetic stirring was applied during the electrolysis to help mitigate the mass transport limitation of dissolved CO_2 in the electrolyte. With an increase...

[Figure. The results of testing multiple Pd and PdNi electrodes at $-0.9 V_{\text{RHE}}$]

2.5. Again for the sake of comparison across the literature, the designs of the electrochemical cell and of the *in-situ* cells should be added in the SI. Furthermore, it should be demonstrated that the conditions in the *in-situ* cells are the same of the electrochemical cells so to actually lead to the same product distribution.

Response: Following the reviewer’s suggestion, we added the designs of the electrochemical cell and the *in-situ* cell in the Supplementary Information. In the *in-situ* X-ray analyses, the electrodes with high areal mass loading (*c.a.*, $>8 \text{ mg cm}^{-2}$) were adopted in order to detect the X-ray signal from the electrode immersed in the electrolyte, where the cell may bring different catalytic activity mainly due to the thick electrodes, although we believe the trend obtained among the various bimetallic catalysts should remain the same. Moreover, the physicochemical information of the samples should be the same in both cells because the diffusion of proton, involved in the phase transition of Pd-to-PdH, would be fast.

Action: To reflect the reviewer’s question, we have added the designs of the electrochemical cell and *in-situ* cell in Supplementary Information.

[4. *In-situ* Measurement on page 16]

The lab-made acryl kit was used for the *in-situ* X-ray measurements (Supplementary Fig. S25). The potential range used for the *in-situ* X-ray measurements was determined after confirming the potential range sufficient for transforming Pd into the PdH phase. *In-situ* XAFS measurements...

[Supplementary Information]

Supplementary Figure 25. The digital photo images for (A) The electrochemical cell and (B) *in-situ* cell used in this study.

2.6. It is noted that the potential reached in the *in-situ* experiments is not the same of the electrochemical measurements. Why is this the case?

Response: In the *in-situ* experiments, the potential range was first determined after confirming the potential range sufficient for transforming Pd into the PdH phase. Our *in-situ* X-ray experiments revealed that bimetallic PdM except for PdPt completed the phase transformation into the PdH phase under the CO₂RR conditions before reaching the potential of -0.4 V_{RHE}. Therefore, we did not need to carry out the *in-situ* experiments at more negative potentials. Moreover, at lower potentials than -0.6 V_{RHE}, it would be difficult to obtain high quality EXAFS data due to the significant amount of gas generation.

Action: To reflect the reviewer's question, we have revised the manuscript as below:

[4. *In-situ* Measurement on page 16]

The lab-made acryl kit was used for the *in-situ* X-ray measurements (Supplementary Fig. S25). The potential range used for the *in-situ* X-ray measurements was determined after confirming the potential range sufficient for transforming Pd into the PdH phase. *In-situ* XAFS measurements...

2.7. The DFT calculations were performed only on (111) surfaces. The authors write "The particle size for all samples is in the range from 5 to 10 nm (Supplementary Fig. 13), thus enabling us to use their (111) surface as a platform for further calculations." It is not clear why this is the case. In fact, spherical nanoparticles are most likely to expose all the facets on their surface. See for example *JACS* 2014, 136, 6978. Can the authors truly justify their statement (perhaps with high res TEM) or otherwise repeat their calculations also on (100) and (110)?

Response: We agree with the reviewer that nanoparticles most likely expose various low index facets such as (111), (100) and (110). We choose the (111) surface as a representative surface of a nanoparticle in DFT modeling as the (111) surface is the thermodynamically most stable low index surface of face centered cubic crystals and is thus expected to be a dominant surface on a nanoparticle. Following the reviewer's suggestions, we performed additional DFT calculations to calculate the *H, *CO and *HOCO binding energies on PdH(100) and (PdNi)H(100) surfaces. The (PdNi)H(100) surface was chosen to represent a bimetallic (PdM)H(100) surface. The DFT calculated binding energies are listed in **Supplementary Table 15**.

The binding energy difference (*i.e.*, BE(*H)-BE(*HOCO)), a descriptor identified in the present study, is larger for (PdNi)H(100) compared to PdH(100). This trend is similar to the trend we observed on the (111) surfaces (see **Supplementary Table 12**). Thus it is expected that the DFT calculated trend on the (111) surface should hold true on the (100) surface. Thus, the (111) surface used in DFT modeling is a reasonable compromise to model surfaces of relatively large nanoparticles, consistent with the practice in many DFT calculations in the literature in identifying trends among different catalysts.

Action: We have revised the main text and Supplementary Information as below:

[Page 12-13]

Hence, the CO₂RR would be accelerated... due to stabilized *H binding and an enhanced HER and/or destabilized *HOCO adsorption. Additional DFT calculations performed on the (100) surfaces of PdH and (PdNi)H show a similar trend in BE(*H)–BE(*HOCO) compared to the corresponding (111) surfaces (Supplementary Table 15). Thus, the (111) surface used in DFT modeling is a reasonable representation in identifying trends of relatively large nanoparticles.

[Supplementary Information]

Supplementary Table 15. DFT calculated binding energies (in eV) of *H, *CO and *HOCO on PdH(100) and (PdNi)H(100) surfaces.

Intermediates	*H			*CO			*HOCO		BE(*H) - BE(*HOCO) (M sites) ^{a)}
	bridge	top-Pd	top-M	bridge	top-Pd	top-M	top-Pd	top-M	
PdH(100)	0.01	0.29	-	-1.16	-1.14	-	-2.07	-	2.36
(PdNi)H(100)	-0.60	-0.08	-0.26	moved to Ni site	-1.55	-2.22	-2.81	-2.86	2.60

^{a)} The BE(*H) at the top site was used to calculate BE(*H)-BE(*HOCO).

2.8. Can the theory or the experiments illustrate how the trends will change with composition within the same alloy?

Response: Following the reviewer's suggestion, we synthesized additional PdNi samples with different bimetallic ratios (denoted as Pd₁₀₀, Pd₈₀Ni₂₀, Pd₇₅Ni₂₅, and Pd₅₀Ni₅₀), followed by the same GC measurements at -0.9 V_{RHE} (**Supplementary Fig. 1** in the revised version). With increasing Ni amount up to 25%, both FE(CO) and J(CO) are better than those of Pd₁₀₀. However, both values decrease at higher Ni contents, as shown in the figure below.

Action: We performed what the reviewer requested and confirmed the trends of J(CO) and FE(CO) with different ratios of Pd and Ni.

[Page 4]

The second metals...an atomic Pd:M ratio of 8:2. This ratio was chosen in this study because PdNi in this ratio (*i.e.*, Pd₈₀Ni₂₀) represented optimal composition for the PdNi catalysts with different ratios (Supplementary Fig. 1). Each final sample was named as PdM...

[Supplementary Information]

Supplementary Figure 1. The trends of J(CO) and FE(CO) at $-0.9 V_{\text{RHE}}$ with different bimetallic ratios of Pd and Ni.

2.9. How does the ECSA from CO stripping compared with the ECSA from double-layer capacitance measurements which is usually employed for Cu-based electrocatalysts? The actual ECSA values should be reported in a table.

Response: The electrochemical methods for ECSA evaluations mostly utilize CO-stripping, surface oxide reduction, and electrical double layer capacitance. Among them, ECSA based on electrical double layer capacitance (denoted as ECSA-EDLC) is usually adopted for those catalysts (*i.e.*, Cu) that show negligible Faradaic reactions over wide potential ranges. This is why ECSA based on CO-stripping (denoted as ECSA-CO) is common in Pt/Pd-based catalysts whose Faradaic reactions, such as H-adsorption and surface oxide formation, take place at several potential regimes. Following the reviewer's suggestion, we have tried to evaluate the ECSA-EDLC values, but it was difficult to define the non-Faradaic potential regime. Instead, we compared the ECSA-CO values with the ECSA from the surface oxide reduction (denoted as ECSA-Pd(OH)₂), which are tabulated in **Supplementary Table 13**. All of the Pd-M bimetallic catalysts show lower ECSA-Pd(OH)₂ values than those of monometallic one, which is probably attributed to the fact that the second metals are not converted into hydroxides. Therefore, the ECSA-CO values are believed to be more meaningful, so that we would like to keep using the ECSA values from CO-stripping.

Action: The ECSA was evaluated from the surface oxide reduction, which is now compared with the ECSA from CO-stripping in **Supplementary Table 13**.

[Electrochemical Measurements on page 15-16]

The charge densities for CO stripping were assumed to be $420 \mu\text{C cm}^{-2}$. For comparison, the values of ECSA using the reduction capacitance ($430 \mu\text{C cm}^{-2}$) of surface Pd(OH)₂ were calculated based on a previous report.⁵¹ The calculated ECSA values are tabulated in Supplementary Table 13.

[Supplementary Information]

Supplementary Table 13. The ECSA values determined from CO-stripping (ECSA-CO) and Pd(OH)₂ reduction (ECSA-Pd(OH)₂) capacitances for the samples.

Entry	ECSA-CO (cm ²)	ECSA-Pd(OH) ₂ (cm ²)
PdAg	10.48949	2.283727
PdCu	10.89659	3.16822
PdNi	8.357829	2.72392
Pd	7.483946	4.47054
PdCo	7.89056	3.68605
PdPt	8.38568	2.79072

Note that all of the ECSA values in the main text are on the basis of ones from the CO stripping method because this is more widely adopted for Pd- and Pt-based catalysts.

2.10. Finally, the standard deviation should be calculated on multiple samples not on multiple GC measurement on the same sample. Values should be corrected.

Response: Following the reviewer's suggestion, we performed several single GC measurements on multiple samples of Pd and PdNi at -0.9 V_{RHE}. As shown in the Figure below, these measurements gave rise to similar results as in the original values.

Action: We performed what the reviewer requested and confirmed the similar error bars as in the original measurements as below:

[Figure]

Reviewer #3 (Remarks to the Author):

In this paper, electrocatalytic CO₂RR and HER using M doped PdH catalysts (M = Co, Cu, Ni, Ag, Pd, and Pt) were studied both experimentally and computationally. Especially, the discussion regarding the correlation between experimental J(CO) and computational $\Delta G(*H)$ - $\Delta G(*HOCO)$ values is interesting. However, there still are some points that need to be clarified as listed below. Once these points are addressed I can recommend publication of this paper in Nature Communications.

Response: We thank the reviewer for his/her thorough review of our manuscript and helpful comments.

1. In this study, H* and HOCO* were assumed to adsorb on M rather than Pd. Depending on M, the adsorption site should change. I therefore recommend to discuss $\Delta G(*H/*HOCO)$ for adsorption on Pd sites.

Response: This question is similar to Comment 4.4 from Reviewer #1. Please see our Response and Action described earlier.

2. The statistical analysis needs to be performed. The authors stated that there is a good correlation between J(CO) and $\Delta G(*H)$ - $\Delta G(*HOCO)$ and $\Delta G(*H)$ - $\Delta G(*HOCO)$ can be a good descriptor for this reaction. However, there is no quantitative evidence showing that there is a good correlation between J(CO) and $\Delta G(*H)$ - $\Delta G(*HOCO)$. I see some correlation also between J(CO) and $\Delta G(*HOCO)$. I also see some correlation between J(CO) and $\Delta G(*H)$ - $\Delta G(*CO)$. How did the authors judge that there is a good correlation or not? Just from their impression? Only from these figures, it is not clear how the authors found correlations among various combinations of experimental and computational values. I recommend to analyze correlations more quantitatively.

Response: Following the reviewer's suggestion, we performed the statistical analysis on the linearity consideration on **Supplementary Figure 18, 20, 22 and 23**. In the scale consideration between J_{ECSA} (from experimental results) and ΔG (from DFT calculation), we first excluded PdPt because it did not lead to hydride formation. Second, in the cases of correlating ΔG of each reaction intermediate and the corresponding J_{ECSA} values, the linear scaling between $\Delta G(*H)$ and J_{ECSA}(H₂) was found. Finally, in order to find out a descriptor that is able to describe both CO₂RR and HER, parameters combining both CO₂RR and HER (*i.e.*, $\Delta G(*H)$ - $\Delta G(*HOCO)$ or $\Delta G(*H)$ - $\Delta G(*CO)$) were introduced. In this regard, even though the correlation between J_{ECSA}(CO) and $\Delta G(*H)$ - $\Delta G(*CO)$ appear to be plausible, $\Delta G(*H)$ - $\Delta G(*CO)$ could not give rise to a high linearity with J_{ECSA}(H₂). Therefore, we reach a conclusion that the value of $\Delta G(*H)$ - $\Delta G(*HOCO)$ is the most reasonable descriptor for explaining both CO₂RR and HER.

We admit that our statistic parameters (R^2) are not that high (between 0.67 and 0.84). However, this is not completely unexpected in the cases of combined experimental and theoretical studies (See figures below from published literature), which might originate from the irregular particle size, surface roughness, and consequently the different catalytic activities in CO₂RR and HER.

[Redacted]

Figure. The examples for correlation of electrochemical activity (J) with the calculated energy from other publications. (A) Electroreduction of CO₂-to-CO with different metals (Feaster *et al.*, *ACS Catal.*, 2017, 7, 4822). (B) Electroreduction of CO₂-to-CO/methane/methanol with different metals (Kuhl *et al.*, *JACS*, 2014, 136, 14107), (C) Oxygen reduction reaction (ORR) with different bimetallic Pt (Stamenkovic *et al.*, *Angew. Chem. Int. Ed.*, 2006, 45, 2897).

Action: We performed the statistical analysis, and the results are now in the Supplementary Figures.

[Supplementary Information]

3. The Statistical Analysis on the Linearity Correlation.

The statistical analysis was performed for the linear correlation of $J_{\text{ECSA}}(\text{CO}/\text{H}_2)$ with the calculated ΔG . In the scale consideration between J_{ECSA} (from experimental results) and ΔG (from DFT calculation), we first excluded PdPt because it did not lead to hydride formation. Second, in the cases of correlating ΔG of each reaction intermediate and the corresponding J_{ECSA} values, the linear scaling between $\Delta G(*\text{H})$ and $J_{\text{ECSA}}(\text{H}_2)$ was found (Supplementary Fig. 20). Finally, in order to find out a descriptor that is able to describe both CO₂RR and HER, parameters combining both CO₂RR and HER (*i.e.*, $\Delta G(*\text{H})-\Delta G(*\text{HOCO})$ or $\Delta G(*\text{H})-\Delta G(*\text{CO})$) were introduced. In this regard, even though the correlation between $J_{\text{ECSA}}(\text{CO})$ and $\Delta G(*\text{H})-\Delta G(*\text{CO})$ appear to be plausible (Supplementary Fig. 22 and 23), $\Delta G(*\text{H})-\Delta G(*\text{CO})$ cannot be a potential descriptor for CO₂RR because it cannot give rise to a high linearity with $J_{\text{ECSA}}(\text{H}_2)$. Therefore, we reach a conclusion that the value of $\Delta G(*\text{H})-\Delta G(*\text{HOCO})$ is the most reasonable descriptor for explaining both CO₂RR and HER.

[Supplementary Information]

Supplementary Figure 18. The correlation of (A) binding energies and (B) ΔG between adsorbed $*H$ and $*HOCO$.

[Supplementary Information]

Supplementary Figure 20. Correlations between free energies of each reaction intermediate and the corresponding J_{ECSA} at (A-C) $-0.9 V_{RHE}$ and (D-F) $-1.0 V_{RHE}$.

[Supplementary Information]

Supplementary Figure 22. Correlations between $J_{\text{ECSA}}(\text{CO})$ at $-0.9 \text{ V}_{\text{RHE}}$ and the free energy difference of (A) $*\text{H}$ and $*\text{HOCO}$ and (B) $*\text{H}$ and $*\text{CO}$. (C-D) The same correlation constructed by using $J_{\text{ECSA}}(\text{H}_2)$ at $-0.9 \text{ V}_{\text{RHE}}$.

[Supplementary Information]

Supplementary Figure 23. Correlations between $J_{\text{ECSA}}(\text{CO})$ at $-1.0 \text{ V}_{\text{RHE}}$ and the free energy difference of (A) $*\text{H}$ and $*\text{HOCO}$ and (B) $*\text{H}$ and $*\text{CO}$. (C-D) The same correlation constructed by using $J_{\text{ECSA}}(\text{H}_2)$ at $-1.0 \text{ V}_{\text{RHE}}$.

3. In this study, only six M were considered. Among the six, Pt doesn't show a good correlation. I suppose that there should be many other cases where the $\Delta G(*\text{H})-\Delta G(*\text{HOCO})$ value is not a good descriptor. I suggest the authors to discuss applicability of the descriptor.

Response: We would like to first state the reason why PdPt does not show a good correlation. This is because it does not lead to hydride formation, so it needs to be considered separately

rather than as a part of the bimetallic PdM catalysts especially in considering the scale correlation. Should PdPt be transformed to (PdPt)H, it would exhibit a good correlation.

The CO₂RR may occur *via* various pathways to produce a wide range of products. It has been shown that the CO production primarily occurs *via* the carboxylic *HOCO intermediate (Sun *et al.*, *J. Am. Chem. Soc.* 2013, 135, 16833) while the formation of formate (*HCOO) intermediate leads to the formation of formic acid as a final product (Sargent *et al.*, *Joule*, Volume 1, Issue 4, 20 December 2017, Pages 794-805). Therefore, the DFT calculations in the present study were carried out to study CO₂ conversion to CO *via* the *HOCO intermediate. Along this reaction pathway, *HOCO and *CO are two key intermediates for the formation of CO. This provides a natural choice of using the binding energy of *HOCO and/or *CO as a potential descriptor of CO₂RR. However, our calculated binding energies of *HOCO and *CO do not correlate well with the experimental selectivity among various (PdM)H catalysts. Thus, we conclude that *HOCO and *CO binding energies alone may not serve as descriptors of CO₂RR on the (PdM)H catalysts. In contrast, we find that the binding energy difference between *H (key intermediate in HER) and *HOCO (key intermediate in CO₂RR), or the $\Delta G(*H) - \Delta G(*HOCO)$, correlates well with the experimentally observed selectivity. Thus, we propose $BE(*H) - BE(*HOCO)$ or $\Delta G(*H) - \Delta G(*HOCO)$ as a potential descriptor of selectivity for the (PdM)H catalysts.

Action: To reflect the discussion above, we added the relevant sentences as below:

[Page 12]

...However, the same consideration using *H or *CO adsorption alone does not correlate with $J_{ECSA}(CO)$ or $J_{ECSA}(H_2)$ as displayed in Supplementary Fig. 22 and Supplementary Fig. 23. The binding energy of O has been predicted... suggesting that the binding energy of O/OH is not a descriptor for CO₂RR over the (PdM)H catalysts. Similarly, a previous study⁴⁸ did not observe a consistent trend between $\Delta G(*H) - \Delta G(*CO)$ and $J(CO)/J(H_2)$ of Cu-deposited Au. Thus, the combined experimental observations and DFT calculations...

Reviewers' comments:

Reviewer #1 (Remarks to the Author):

According to the revised manuscript, the authors have indeed resolved some of issues. However, some issues have remained unsolved. I still think that the authors oversell their results.

First of all, the authors found that there is minor formic acid products in the supplementary experiment, then why not consider the HCOO^* mechanism or the adsorption energy of HCOO^* ?

Secondly, is the reaction under diffusion control or chemical control with stirred? From the revised manuscript, it seems that the rate-limiting step is still mass transport of dissolved CO_2 , in which case the current density associated with CO_2 conversion should not change with the catalyst. The author needs to give a clear judgment at least some experimental evidence.

Thirdly, XAFS should preferably provide a C spectrum or an O spectrum to give information about the intermediate in mechanism research experiments.

Fourthly, it should be helpful using isotope experiments when studying the role of PdH.

Finally, the authors found that the selectivity depend on different M rather than Pd. What's the intrinsic reason? Can In-situ XAFS select the active site between Pd and second M?

Reviewer #2 (Remarks to the Author):

The authors have addressed most of my concerns. However the morphological/compositional characterization of the catalysts still remains unsatisfactory. Performing STEM-EDX mapping on just one single particle (Figure S10) is certainly not enough. Also quantification of the two metal ratio for both this sample and the new added samples (Figure S1) is really needed. If elemental analysis by ICP-OES or ICP-MS is not available, can the authors use the EDX data. The manuscript will be ready for acceptance after these final additions.

Reviewer #3 (Remarks to the Author):

I found that the authors addressed all my points in their response. I can recommend publication of this paper in Nat. Comm.

Response to Reviewers' Comments:

Reviewer #1 (Remarks to the Author):

According to the revised manuscript, the authors have indeed resolved some of issues. However, some issues have remained unsolved. I still think that the authors oversell their results.

Response: We thank the reviewer for his/her thorough review of our manuscript and helpful comments.

1. First of all, the authors found that there is minor formic acid products in the supplementary experiment, then why not consider the HCOO* mechanism or the adsorption energy of HCOO*?

Response: As stated in the revised manuscript, this study focuses on the syngas production with tunable CO/H₂ ratios. This is because the yield of formic acid is low when compared with those of CO and H₂ formation. Especially, the formation of formic acid is negligible at -0.9 V_{RHE} and thereafter, where our main scientific effort is focused (**Figure 4F** and **Supplementary Figure 20-24**). Therefore, the detailed reaction pathway for formic acid is not considered in this manuscript.

Action: We have revised the manuscript as below:

[Page 11]

...CO₂RR intermediates (*i.e.* *HOCO and *CO) over PdH and (PdM)H surfaces (Supplementary Table 12). In the DFT calculation for CO₂RR, the reaction pathway for CO production was only considered because of the negligible yield of HCOOH formation in the experiments. The particle size for all samples...

2. Secondly, is the reaction under diffusion control or chemical control with stirred? From the revised manuscript, it seems that the rate-limiting step is still mass transport of dissolved CO₂, in which case the current density associated with CO₂ conversion should not change with the catalyst. The author needs to give a clear judgment at least some experimental evidence.

Response: As displayed in the Faradaic efficiency (FE) profiles (**Supplementary Figure 4**), all of the samples show the same tendency that FE(CO) initially increases, then saturates, and finally decreases with increasing applied overpotentials. This indicates that CO₂RR is limited by the mass transport of CO₂ gas onto the catalysts at high potentials, which is consistent with the previous literatures on CO₂RR. As for the J(CO) and J(H₂) with different catalysts, their values indeed change in Pd-M bimetallic catalysts with different M (**Figure 1C** and **Supplementary Figure 7**), which is an indication of the tuned binding affinity of the reaction intermediates (*HOCO, *CO, and *H) involved in CO₂RR and HER. This is why we could compare the trends in J(CO) values of different Pd-M bimetallic catalysts at different potentials.

Action: We have revised the manuscript as below:

[Page 5]

...negligible at $-0.9 V_{\text{RHE}}$ and thereafter. As the potential is scanned more cathodically, FE(CO) tends to **initially increase, then saturate, and finally decrease** while FE(H₂) increases. This is because CO₂RR is controlled by...

3. Thirdly, XAFS should preferably provide a C spectrum or an O spectrum to give information about the intermediate in mechanism research experiments.

Response: *In-situ* XAFS study in this work utilizes the high energy X-ray photons of > 10,000 eV, which is not sensitive to the low energy elements such C and O. The C (~270 eV) and O K-edge (~530 eV) features are in the vacuum ultraviolet (VUV) regions. Photons in this energy range requires vacuum environment, making it extremely difficult (if not completely impossible) to perform *in-situ* measurements in an operating electrochemical cell like the one used in the current study.

4. Fourthly, it should be helpful using isotope experiments when studying the role of PdH.

Response: While isotope experiment using deuterium (D⁺) instead of proton (H⁺) might be useful in understanding the structure of PdD (or PdH), this experiment is not likely able to provide anything about the impact of PdH formation on the syngas production. In fact, the role of PdH formation on CO₂RR is already described on pages 5, 8, and 9 of the revised manuscript; Pd produce hardly any CO without PdH formation due to the strong binding affinity of CO. Moreover, in considering our CO₂RR condition where the constant potential is applied, both PdH and PdD will be the active phase for the given reaction. Therefore, we kindly ask the reviewer to understand this point.

5. Finally, the authors found that the selectivity depend on different M rather than Pd. What's the intrinsic reason? Can In-situ XAFS select the active site between Pd and second M?

Response: The results obtained from the experimental measurements and DFT calculations show that the selectivity of the Pd-M based catalysts in the current study is modified by the presence of the second metal (M). Additionally, DFT calculations show that the binding energies/free energies calculated on the M sites correlate well with the experimental results. Thus, as described in the manuscript, the intrinsic reason is that binding energy/free energy calculated on the M sites as a potential descriptor even though the difference in selectivity of PdMH catalysts originates from the change in electronic structure of surface catalytic sites (Pd and/or M) due to alloying between Pd and M.

Reviewer #2 (Remarks to the Author):

The authors have addressed most of my concerns. However, the morphological/compositional characterization of the catalysts still remains unsatisfactory. Performing STEM-EDX mapping on just one single particle (Figure S10) is certainly not enough. Also quantification of the two metal ratio for both this sample and the new added samples (Figure S1) is really needed. If elemental analysis by ICP-OES or ICP-MS is not available, can the authors use the EDX data.

The manuscript will be ready for acceptance after these final additions.

Response: As for STEM-EDX mapping, additional images after CO₂RR are now included (**Supplementary Figure 10**). The elemental ratio of Pd and M determined by ICP-OES is also included for all of the samples in the revised manuscript. The results are tabulated in **Supplementary Table 1**. We thank the reviewer for the constructive review of our manuscript and helpful comments.

Action: To reflect the reviewer's request, we have revised as below:

[Page 4]

...at an atomic Pd:M ratio of 8:2. This ratio was chosen in this study because PdNi in this ratio (*i.e.*, Pd₈₀Ni₂₀) represented optimal composition for the PdNi catalysts with different ratios (Supplementary Fig. 1 and Supplementary Table 1). Each final sample...

[Page 14]

...Then, the final product of 10wt% Pd-based bimetallic NPs on C was obtained and denoted as PdM for the bimetallic cases. Inductively-coupled plasma-optical emission spectroscopy (ICP-OES, Optima 8300, PerkinElmer) confirmed the successful synthesis of bimetallic PdM with desired Pd/M atomic ratios (See Supplementary Table 1). For the Au/C sample, gold chloride (AuCl₃)...

[Supplementary Information]

Supplementary Table 1. The elemental analysis by ICP-OES for the bimetallic PdM samples in this study.

Entry	Pd (ppm)	M (ppm)	mol ratio of Pd/M ^{a)}	x in Pd _{100-x} M _x
Pd ₈₀ Ag ₂₀	0.428	0.105	4.13	19.5
Pd ₈₀ Cu ₂₀	1.308	0.201	3.89	20.5
Pd ₈₀ Ni ₂₀	1.460	0.205	3.93	20.3
Pd ₇₅ Ni ₂₅	1.146	0.254	2.49	28.7
Pd ₅₀ Ni ₅₀	1.113	0.534	1.15	46.5
Pd ₈₀ Co ₂₀	0.776	0.121	3.55	22.0
Pd ₈₀ Pt ₂₀	0.957	0.467	3.76	21.0

^{a)} Pd (106.42 g mol⁻¹), Ag (107.868 g mol⁻¹), Cu (63.546 g mol⁻¹), Ni (58.693 g mol⁻¹), Co (58.933 g mol⁻¹), Pt (195.084 g mol⁻¹).

[Supplementary Information]

Supplementary Figure 10. STEM-EDX mapping images of Pd (red) and Ni (green) before and after CO₂RR.

Reviewer #3 (Remarks to the Author):

I found that the authors addressed all my points in their response. I can recommend publication of this paper in Nat. Comm.

Response: We thank the reviewer for his/her thorough review of our manuscript and helpful comments.

REVIEWERS' COMMENTS:

Reviewer #1 (Remarks to the Author):

The authors try their best to answer my concerns. However, the authors' response is still unsatisfactory, because no convincing experimental evidences have been provided. Of course, I would like to leave this to the editor to make the final decision.

Reviewer #2 (Remarks to the Author):

The manuscript is ready for acceptance

Response to Reviewers' Comments:

Reviewer #1 (Remarks to the Author):

The authors try their best to answer my concerns. However, the authors' response is still unsatisfactory, because no convincing experimental evidences have been provided. Of course, I would like to leave this to the editor to make the final decision.

Response: To reflect the reviewer's previous comments on the pathway of formic acid (HCOOH) formation *via* the *HCOO intermediate, we have calculated the difference in the Gibbs free energy change $\Delta(\Delta G)$ between the first steps of CO (*HOCO) and HCOOH (*HCOO) formation over several bimetallic systems (*e.g.*, PdAg, PdCu, PdNi, and Pd).

Based on the new DFT results, for those catalysts that are favorable for CO production, the formation of the *HCOO intermediate is slightly favored compared to *HOCO. However, considering the low yield of formic acid at high overpotentials, the new DFT results suggest that the production of CO could also be potentially promoted from the *HCOO pathway. While more detailed study using *in-situ* infrared and Raman spectroscopies will be needed to further characterize the surface intermediates, our gas phase DFT calculations (without solvation) indicate that both *HCOO and *HOCO intermediates potentially lead to CO production.

Action: To address the reviewer's comment, we have revised the manuscript as below:

[Page 11] Remove the highlighted sentence.

...PdH and (PdM)H surfaces (Supplementary Table 12). ~~In the DFT calculation for CO₂RR, the reaction pathway for CO production was only considered because of the negligible yield of HCOOH formation in the experiments.~~ The particle size for all samples...

[Page 12-13] Add the highlighted sentence.

...due to stabilized *H binding and an enhanced HER and/or destabilized *HOCO adsorption. The ΔG values of the *HCOO species, which is a key intermediate for HCOOH formation, over PdH and (PdM)H were calculated (Supplementary Table 16). For those catalysts that are favorable for CO production, the formation of the *HCOO intermediate is slightly favored over *HOCO. However, considering the low yield of formic acid at high overpotentials, the DFT results suggest that the production of CO could also be potentially promoted from the *HCOO pathway. While more detailed study using *in-situ* infrared and Raman spectroscopies will be needed to further characterize the surface intermediates, the DFT results in Supplementary Table 16 suggest that both *HCOO and *HOCO intermediates potentially lead to CO production. Additional DFT calculations performed on the (100) surfaces...

[Supplementary Information]

Supplementary Table 16. DFT calculated difference in the Gibbs free energy change ($\Delta(\Delta G)$), in eV) between *HOCO and *HCOO intermediates over PdH and (PdM)H.

Entry	$\Delta(\Delta G)^a$
-------	----------------------

(PdAg)H	0.20 ^{b)}
(PdCu)H	0.08 ^{b)}
(PdNi)H	0.00
PdH	-0.24

a) Negative $\Delta(\Delta G)$ values indicate the formation of *HCOO is favorable over that of *HOCO.

b) Even though the *HCOO intermediate is slightly favored, the production of formic acid was negligible at high overpotentials, suggesting that both *HOCO and *HCOO pathways can promote CO production.

Reviewer #2 (Remarks to the Author):

The manuscript is ready for acceptance.

Response: We thank the reviewer for his/her thorough review of our manuscript and helpful comments.